

# A closer look at IRSL SAR fading data and their implication for luminescence dating

Annette Kadereit[1], Sebastian Kreutzer[2, 3], Christoph Schmidt[4], Regina DeWitt[5]

[1] Heidelberger Lumineszenzlabor, Geographisches Institut, Universität Heidelberg, Im Neuenheimer Feld 348, 69120 Heidelberg, Germany

[2] Geography & Earth Sciences, Aberystwyth University, Aberystwyth, SY23 3DB, United Kingdom

[3] IRAMAT-CRP2A, UMR 5060, CNRS-Université Bordeaux Montaigne, Pessac, France

[4] Lehrstuhl Geomorphologie, Universität Bayreuth, Universitätsstr. 30, 95447 Bayreuth, Germany

[5] Department of Physics, East Carolina University, C-209, Howell Science Complex, 1000 E. 5th Street, Greenville, NC 27858, USA

*Correspondence to*: Annette Kadereit (annette.kadereit@uni-heidelberg.de)

**Graphical abstract**

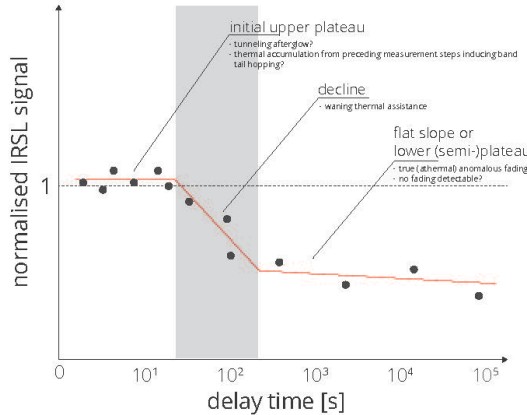

**Highlights**

- IRSL SAR fading data at a much closer spacing than usual in luminescence dating

- High-resolution fading curves revealing unexpected shapes

- Shape of fading curves not consistent with the model of logarithmic signal decline

- Curve shapes varying and dependent on SAR measurement parameters

- Results admonishing for caution in calculating fading rates and fading corrected IRSL ages

**Abstract.** Feldspar, used for infrared stimulated luminescence (IRSL) dating, is known as a dosimeter which might not completely retain the environmental dose over time, therefore leading to age underestimation. The dose leakage is believed to

be caused by non-thermal (anomalous) charge redistribution in the crystal and reflected in an IRSL signal which diminishes with time accordingly. After laboratory irradiation, this signal decline may be monitored by successive IRSL readouts following increasing delay times. Hence, tests of anomalous signal fading are integral steps of IRSL dating procedures applied to feldspar and feldspar-bearing polyminerals. In these measurements IR-stimulation is in most cases preceded by thermal pretreatment (preheating) of the sample. Per common practise, preheating is performed immediately after laboratory irradiation

to avoid unwanted electron redistribution assumed to occur if preheating is performed immediately before the delayed IRSL-





readout. Here we compile a series of single aliquot regeneration (SAR) measurements questioning this practice. As a result, the fading measurements may possibly reveal post-irradiation afterglow. The results also suggest that data curves resembling anomalous fading may be caused by insufficient control of the readout temperature waning with increasing delay time. The unwanted effects are observed best for IRSL at room-temperature and on luminescence readers with an out-of-date steering

software, but they are relevant also for IRSL at elevated temperature and on modern readers, likely including novel post-IR IRSL (pIRIR) protocols. For temperatures as homogeneous as possible during IRSL readout of the (fading) dose, we recommend preheating immediately prior to (delayed) IRSL-readout in order to avoid measurement artefacts either resembling entirely anomalous fading of the IRSL-signal or increasing the true values. It should be noticed that multifold SAR protocol and measurement parameters, like e.g. the type of luminescence reader or the use of $N_2$ flow, may further modify the course

of the data values and therefore the amount of the measured signal loss in a particular time interval after laboratory irradiation. Furthermore, calculations of signal fading ($g$-value) should consider only IRSL-readout after a minimum delay time after laboratory irradiation to avoid including possible post-irradiation afterglow in $g$-value determination. The measurements compiled in the present study were performed on polymineral fine grains extracted from loess-borne samples from southern Germany and a limnic sample from Mexico. Therefore, the observations are assumed to be not only of local or regional interest

but they appear to be of general relevance to SAR fading tests. However, with respect to the likely varying temperatures during IRSL readout of the fading dose administered in the laboratory, the observations are at least partly owed to the promptly measured test dose for normalizing the preceding (fading) dose. This is in contrast to classical multiple aliquot additive (MAA) measurements in which preheating may be replaced by long storage of a sample after laboratory irradiation and in which fading tests may be designed to also correct for possibly (slightly) changing IRSL readout temperatures at different delay times. Thus,

the observations are at least partly SAR-immanent.

**Keywords.** polymineral fine grains, infrared stimulated luminescence (IRSL), single aliquot regeneration (SAR) protocol, anomalous fading

## 1    Introduction

### 1.1    General considerations

Feldspar and feldspar-bearing mineral separates extracted from sediments, ceramics or archaeological structures are widely used chronometers in optically and infrared stimulated luminescence (OSL, IRSL) dating (e.g., Aitken 1998). However, since the study of Wintle (1973) on feldspar-bearing lava, the malign phenomenon of anomalous fading became associated with feldspar and feldspar-bearing mineral separates. The term "fading" refers to the observation that the mineral's luminescence signal does not completely represent the accumulated paleodose. In general, at a given storage temperature, electrons trapped

at defects in the feldspar crystal (IRSL trap), have a certain probability to overcome the energy difference between the IRSL trap and the conduction band, which for feldspar is ~ 2 eV (Huett et al. 1988). "Anomalous", however, means that the observed decrease of the luminescence signal cannot be explained exclusively by thermally assisted escape of trapped electrons. Several mechanisms have been proposed to explain anomalous fading: Visocekas (1985; 1993), Visocekas et al. (1994; 1998) and Spooner (1994) suggested charge recombination caused by quantum-mechanical tunneling as source of this effect, while

Templer (1986) additionally introduced localized transitions as a complementary process. Tyler and McKeever (1989) concluded for their experiments on oligoclase that the localized transition model provides a better match for fading signals than the quantum mechanical tunneling model. In general, the localized transiation mechanism appears to dominate above room temperature, while tunneling recombination is more important at lower temperatures (Templer 1986). This view is shared by Wintle (1977) and Molodkov et al. (2007) who dinstinguish between temperature-independent tunneling and a temperature-

dependent mode of fading. This second temperature-dependent fading component was later related to a "hopping" mechanism through the band tail states of feldspar (Guérin and Visocekas, 2015), which occurs particularly intensely in highly disordered



volcanic feldspar. Accordingly, these authors conclude that due to this temperature-dependent fading mode, fading measurements in the laboratory at room temperature do not necessarily represent the thermal history experienced by a sample in nature. In addition to quantum-mechanical tunneling from the ground state and/or the excited state of the dosimetric trap, in their comprehensive feldspar model Jain and Ankjærgaard (2011) proposed thermally assisted (0.05–0.06 eV) electronic diffusion in band-tail states as a mechanism for fading, alongside quantum-mechanical tunneling from deep, thermally disconnected band tail states.

Signal loss associated with anomalous fading follows logarithmic decay over time (Visocekas 1985; Zink 2008). This means that, following irradiation in the laboratory or in nature, the absolute rates of signal loss are high in the beginning (with traps being relatively strongly occupied and centers being least occupied) and become smaller over time (with traps emptying and centers being depleted successively), but never reach zero. The higher rate of signal loss in the beginning facilitates the observation of anomalous fading in the laboratory on artificially irradiated samples. However, deviations from the simple power-law signal decay occur in case the irradiation time of the sample is in the same order as the delay between dosing and luminescence measurement (Visocekas 1985; Molodkov et al. 2007).

As anomalous fading would lead to age underestimation, it has become common practice to perform fading tests along with IRSL dating of feldspar and feldspar-bearing fine-grains (e.g., Lamothe et al. 2012). Such fading tests allow detecting IRSL-signal loss, at least on laboratory time scales (short-term fading). Recently, it was shown that the size of fading rates is also relevant for extracting a stable signal from thermoluminescence (TL) curves for low-temperature thermochronology-dating of bedrocks (Brown & Rhodes 2019).

## 1.2    Methodical and technical implementation

To determine the natural dose (N) accumulated by a sample, the luminescence signal of the natural sample is compared to the luminescence signal induced by a calibrated laboratory source (usually a $^{90}Sr/^{90}Y$ beta source). Several laboratory doses (LAB) are used to build up a dose-response curve which describes the dependency of the size of the luminescence signal on the laboratory dose. This may be done with a multiple aliquot additive (MAA) approach, in which LAB is administered on top of a sample's natural dose (N + LAB) or with a regenerative approach, in which LAB is administered after N has been depleted (Bleach + LAB). In the currently favoured regenerative approach, the determination of the natural dose occurs successively on the same aliquot on which first the natural luminescence signal ($L_x$) is determined. This modern single aliquot regeneration (SAR) protocol (Murray and Wintle 2000) requires the monitoring of possible sensitivity changes of the aliquot during a complete measurement cycle. The correction, or normalisation, occurs with the help of the luminescence signal ($T_x$) of a repeatedly administered, constant laboratory dose (normalisation dose, NRM). The SAR-corrected luminescence signals ($L_x/T_x$) are used for dose-response curve construction.

Prior to IRSL readout the feldspar sample is thermally treated, a practice called "preheating". Preheating eliminates potentially short-lived electron capture at mineral defects (instable traps) of the artificially irradiated sample and aligning electron charge distribution of the artificially irradiated sample with that of the natural sample. Common preheating procedures for the dating of feldspar and feldspar-bearing separates are 120 s at 220 °C (e.g., Lang & Wagner 1997; Kadereit et al. 2006), 60 s at 250 °C (Auclair et al. 2003; Balescu et al. 2003; Lamothe et al. 2001), 10 s at 280 °C (Li et al. 2008; Gong et al. 2012), and as of late for post-IR IRSL (pIRIR) dating, 60 s at 320 °C (Thiel et al. 2011). However, preheating may alter the luminescence characteristics of feldspars, which was considered to be the actual source of signal fading (Jaek et al. 2007; Molodkov et al. 2007).





### 1.3 Fading tests for SAR-dating

For SAR dating, Auclair et al. (2003) developed a protocol for fading measurements. Basically, following the SAR cycles used for sample dating, further SAR cycles are added, yet with a constant laboratory dose (Bleach + $LAB_{fad}$; with Bleach + $ß_{fad}$ being the corresponding IRSL signal) and increasing delay times (pauses) in between the irradiation and the IRSL-readout of an aliquot. The dependency of the signal decay on delay time is on a logarithmic time scale described by a linear function (cf.

Fig. 4A in Auclair et al. 2003). According to the fading model, anomalous signal fading occurs already during laboratory irradiation. Depending on the strength of the laboratory source and the administered dose, irradiation times range mostly from a few seconds to several minutes, but may also last longer. Therefore, it is not possible to measure the full amount of signal loss. For comparing different samples, the signal decay is normalised to a given time after the laboratory irradiation, or rather to the point in time representing the middle of the irradiation period. With respect to that reference period (e.g., 2 days), the

relative decay may be expressed as loss of signal in percent per decade (so called *g*-value, Aitken 1985). Often, a *g*-value which was obtained from signal loss observed at laboratory time scales is used for correcting the palaeodose (geologic time scales; long-term fading) and for calculating fading-corrected IRSL ages (e.g., Lamothe & Auclair 1999). This shows that accuracy is essential in any *g*-value determination.

### 1.4 Current best-practice rules for SAR-based fading tests

Auclair et al. (2003) have developed rules for measuring anomalous signal fading with the SAR protocol, which are widely accepted (e.g., Zink 2008). Based on observed signal losses for delayed IRSL readout, the authors strongly recommend that preheating shall not be carried out immediately prior to IRSL readout (pause in between laboratory irradiation and preheating), but immediately after laboratory irradiation (pause in between preheating and IRSL-readout). The authors demonstrated that this way of measurement provides larger *g*-values, which were regarded as reliable. In contrast, the smaller *g*-values associated

with preheating immediately prior to IRSL-readout were interpreted to underestimate the true *g*-values. The authors regarded charge redistribution in the feldspar crystal caused by delayed preheating as being responsible for the *g*-value underestimation.

### 1.5 Doubts about the usual practice of SAR-based fading tests

Unlike Auclair et al. (2003), Rhodius et al. (2015) could not use a fully automated luminescence reader, when dating *in situ* feldspar minerals within stone surfaces. They had to perform preheating manually in an external oven, and to cool down the

135 samples to room-temperature (here and in the following *sensu* ambient temperature, surrounding temperature) manually prior to IRSL readout. The cooling occurred immediately after the preheating on a water-cooled copper plate for 300 s for ˜ 20 °C. Further, unlike Auclair et al. (2003) who used an elevated readout-temperature of 50 °C on a Risø reader model TL/OSL DA15 (Bøtter-Jensen et al. 2000), the so called LasLUM reader (Greilich 2004) used by Rhodius et al. (2015) allows IRSL readout only at room-temperature. Similar to Auclair et al. (2003), Rhodius et al. (2015) found larger *g*-values when preheating and

140 cooling the samples immediately after laboratory irradiation, and cooling the samples again prior to the delayed measurement of the IRSL signal (Bleach + $ß_{fad}$). However, when the samples were preheated and cooled down immediately prior to IRSL readout, no fading was observed. For stone-surface dating on the LasLUM reader, Rhodius et al. (2015) considered the latter procedure as the only appropriate one to provide constant temperatures at the sample position during both prompt and delayed IRSL-readout. Had the sample that had been stored for fading at room-temperature not been cooled prior to reading out

Bleach + $ß_{fad}$, the readout-temperature might have been higher than at the stone surfaces that for dating had first been preheated for 150 s at 225 °C and right afterwards cooled down for 300 s at ~ 20 °C. A possibly higher readout temperature during the delayed IRSL readout could falsely lead to too small *g*-values. Yet, cooling a sample that had been stored at room-temperature during the fading period probably could diminish the temperature of the stone below the wanted readout-temperature. This may lead to too large *g*-values. Thus, the study of Rhodius et al. (2015) showed that temperature control may be a crucial issue

for fading tests, at least if IRSL readout occurs at room-temperature and not at elevated temperatures.




### 1.6 Further inconsistencies in the context of SAR based fading tests

For IRSL-readout at an elevated temperature, observations were reported which are not in agreement with the model of logarithmic signal decay. Steffen et al. (2010) found that signal loss after 2 days was not any different from signal loss after 88 or 102 days, thus suggesting that fading came to a standstill after a few days. For a sequence of laboratory doses read out with increasing delay Auclair et al. (2003) observed unexplainable behaviour of the initial measurement point. Using a novel pIRIR approach (Thomsen et al. 2008), Thiel et al. (2011) found fading for samples which yielded ages in agreement with the stratigraphic placement, and consequently considered the *g*-values being measurement artifacts. Trauerstein et al. (2012; 2014) showed that *g*-values determined on single grains (SG) tend to be lower than those gained from single aliquots (SA) of the same sample material. An explanation could not be given for that observation. In many publications, the *g*-value determination is based on only few data points, e.g., on one readout with short delay of several minutes to less than one hour and on two more readouts after longer periods of one and two days, respectively (e.g., Lomax et al. 2014; Trauerstein et al. 2014). Often, there is hardly any difference in the SAR values of the longest and the second longest interval, and the gradient of the regression line is therefore determined much by the difference of the first SAR value *versus* the level of the last two SAR values (cf., e.g., Fig. 6a in Trauerstein et al. 2014 and Fig. 3a in Preusser et al. 2014). Fading-corrected ages calculated with SAR based *g*-values have repeatedly been reported to either overestimate or to underestimate the expected ages (e.g., Li 2018; Lowick et al. 2012; Wallinga et al. 2007). Most remarkable with respect to the present study, loess-borne sediment samples from southwestern Germany which had been dated with an MAA protocol usually did not show any fading, if signal detection was restricted to the blue-violet (410 nm) emission band (e.g., Lang & Wagner 1996; Kadereit 2002). For details of the MAA fading tests see supplement 1. When, however, fading was measured with a SAR protocol signal loss was observed (cf. section 3).

### 1.7 Scope of the present study

The differing results of the fading tests of the MAA and the SAR approaches were surprising. Therefore, SAR fading tests on polymineral fine grains were investigated in more detail in the present study. The presentation of the results starts with SAR fading tests performed with IRSL-readout at room temperature, because the MAA fading tests had been performed that way and because relevant effects may very well be illustrated based on these (cf. section 3.1). The test series will be extended to samples read out at elevated temperatures showing that for reliable fading tests temperature control is relevant also for these (cf. sections 3.2, 3.3). It is not intended to produce actual *g*-values, as these would be irrelevant for our study, and could not be transferred to other dating applications. The aim is to illustrate the general effects which may occur in the course of SAR fading tests. The magnitude of the effects, however, depends on the particular measurement setup and protocol (e.g., type of the luminescence reader, IRSL-readout temperature, preheat procedures, number of aliquots measured in one sequence) and therefore would need to be traced and quantified in each luminescence laboratory and for each dating study individually. At first sight, some of the tests presented here might be regarded as not well conceived, e.g., if a higher liftup temperature than IRSL-readout temperature is chosen. The liftup temperature gives an upper limit for the temperature of the heating plate, at which an aliquot can be lifted from the turntable to the measurement position. The results of these tests, however, sensitize for possible interpretations of further tests. Our experiments also include measurements on an older reader type which is still in use (e.g., Goldsmith et al. 2017) and on which inevitably numerous *g*-values were produced in the past.

### 2 Methods and samples

### 2.1 Luminescence readers

IRSL measurements were carried out on two luminescence readers in the Heidelberg Luminescence Laboratory. (1) Risø reader model TL/OSL DA12 (serial number 27; Bøtter-Jensen, 1988, 1997; Bøtter-Jensen et al., 1991) is equiped





with a turntable with 24 sample positions, a ring of TEMT484 diodes for sample stimulation in the near infrared at 880 Δ 80 nm (~ 40 mW cm$^{-2}$ at the sample position), a bialkali photomultiplier tube (PMT) EMI9235Q for the detection of the luminescence signal, a $^{90}$Sr/$^{90}$Y source for β-irradiation (~ 1.7–1.5 Gy min$^{-1}$ in the period of the measurements 2012–2016), and a heating unit for thermal treatment of the samples. The automated measurements were controled by a Risø interface

(hardware) and a Risø software (TL.exe) operated in a DOS-emulation mode (TLM.exe, Risø National Laboratories, MT-Software 1994, Version 4.65). Hardware and software setup were the same that had been used by Lang et al. (2003) and Kadereit et al. (2010).

(2) Risø reader model TL/OSL DA20 (serial number 240, nicknamed "Athenaeum"; DTU Nutech 2020) is equipped with a turntable with 48 sample positions, three clusters of infrared emitting diodes (7 LEDs each; 870 Δ 40 nm) for infrared

stimulated luminescence (IRSL), four clusters of blue light emitting diodes (7 LEDs each; 470 Δ 30 nm) for blue light stimulated luminescence (BLSL), a bialkali PMT EMI 9235QB15 for signal detection, a $^{90}$Sr/$^{90}$Y β-source (~ 5.8 Gy min$^{-1}$ at the time of measurements 2016–2017) for laboratory irradiation, and a heater unit for preheating the samples. The computer (Risø MiniSys) which controls the measurements on the luminescence reader, including the temperature control of the heating unit, was run with the Risø MiniSys software version 4.08 (12.01.2016). Mesurements were run with the Risø sequence editor

v4.36 (2015-09-10).

Signal detection for all IRSL measurements on both readers occurred in the blue-violet spectrum around 410 nm, through an interference filter CH-30D410-50 (Chroma) on model DA12 and CH-30D410-44.3 (Chroma) on model DA20, respectively. The readers are connected to a nitrogen-gas supply-pipe, so that measurements may, or may not, be run with continuous or intermittent nitrogen flow.

**2.2    Measurement protocol**

Fading measurements were performed using the SAR protocol (cf. section 1) applying 240 s IR-stimulation. One to three aliquots were measured in each test. We did not use the option "run one aliquot at a time". Several measurement parameters were varied, e.g., fresh as well as previously measured aliquots were used; IRSL readout occurred at room-temperature or at elevated temperatures (50 °C, 60 °C; mostly after the readout temperature had been stabilized for either 5 s or 10 s); preheat

procedures were 120 s at 220 °C, 60 s at 250 °C, 20 s at 280 °C or 60 s at 280 °C. Also, the number and length of pauses for delayed IRSL-readout were varied, e.g., to investigate in more detail shorter pauses, or to reduce the total measurement time by skipping interjacent pauses. Measurements compiled in the present study occurred (1) with $N_2$ purge only at the beginning of a measurement, (2) without $N_2$ flow, (3) with continuous or (4) intermittent $N_2$ flow. In some tests additional heating was performed on empty turntable positions. Laboratory doses (LAB) and normalisation doses (NRM) were varied, e.g., with

respect to the brightness of the samples. Measurement parameters are given in sections 3.1.1. – 3.3.2 and are compiled in supplement 2.

**2.3    Samples**

The IRSL measurements were performed on the feldspar component of polymineral fine grains (~ 4 – 11 µm) pipetted onto aluminium discs (diameter ~ 10 mm, thickness ~1 mm). The mineral separates were extracted from loess-borne sediment

samples from southwestern Germany (HDS-504, HDS-713; Kadereit et al. 2010), and from a limnic sediment sample from Satillo on Lake Chapala in Mexico (HDS-1712; Kadereit et al. 2017). The loess separates were the comparatively brighter samples (several 10³ counts for the IRSL interval (I) 1 – 20 s for LAB ~ 4.6 Gy and NRM ~ 2.3 Gy; several 10⁴ – 10⁵ counts for LAB ~ 10.3 Gy and several 10⁴ counts for NRM ~ 5.2 Gy), while the limnic separates proved rather dim (few 10³ counts for I 1 – 20 s for LAB and NRM each ~ 41.4 Gy).





**2.4    Data handling and graphical representation of the results**

Initial data handling used the software "Analyst" v4.31.9 (Duller 2015), or for the earlier measurements with a predecessor of the same software by the same author. Further data processing and graphical visualization occurred with Microsoft EXCEL$^{TM}$ 2016 and SigmaPlot v11.0. We present our results in graphs, in which we do not include error bars for the sake of clearness, but which are simple scatter graphs with lines connecting the individual symbols as a guide for the eye (see Figs. 2ff). Further,

for easier visual perception we disregard the slightly differing time intervals for the prompt IRSL readout after laboratory irradiation which in reality increase with an increasing irradiation time (if the mid-point of the irradiation time is taken as the zero point), but denote them uniformly as "0 s" on the x-axes of Figs. 4ff (both at the beginning and at the end of a measurement, i.e. the latter after the respective breach in the x-axis). As the results of the fading tests may show some scatter for any of the different analysed signal intervals (here: 0–10 s, 0–20 s, 0–30 s), which might distract from the general data trend, we present

the results for several intervals, as to better bring out the overall course of the data. The different intervals are presented in the respective figures of the main text (Fig. 2 to Fig. 8) in the same colours: 0 – 10 s in green, 0 – 20 s in blue and 0 – 30 s in red. Since the very early and short intervals (e.g., 0–1 s) may show stronger scatter we did not consider these for graphical display. For the same reason, we decided to present the results of the gross values and not the net signals after late light (LL) subtraction (Aitken & Xie 1992; here 51–60 s), as in individual cases the latter might show stronger scatter. After all, the net signals show

basically the same trends as the gross values (cf. supplement 3). Illustrated are double normalised SAR IRSL-signals, i.e., all SAR-corrected signals ($L_x/T_x$; first normalisation) are further normalised to the first illustrated SAR[1] run of the sequence ($[L_x/T_x]/[L_1/T_1]$). As the logarithm of zero is not defined we moved the promptly read-out dose points (0 s delay time) for graphical presentability on the logarithmic time scale to 0.2 s in Fig. 1c and Figs. 4ff.

**3    Results**

From the series of approximately 40 fading tests (FT) compiled in supplement 2 we selected about half for graphical presentation and discussion in the main text ($T_{fad}$-1 – $T_{fad}$-18). Few further measurements as well as presentations of net values are compiled in supplement 1 and 3. The presentation of results starts with measurements of loess-borne samples from SW-Germany on reader DA12 (section 3.1.1) and continues with measurements on reader DA20 (section 3.1.2 ff). Further, the results are arranged with respect to increasing intensity of the preheat procedures from 120 s at 220 °C (section 3.1), via 60 s

at 250 °C (section 3.2) to 20 s at 280 °C and 60 s at 280 °C (section 3.3). Whereas for the lowest and highest preheat temperatures results from loess-borne samples from SW-Germany were selected, a limnic sample from Lake Chapala in Mexico was chosen for the intermediate preheat temperature. The measurements compiled in sections 3.2 and 3.3 occurred, with the exception of one test ($T_{fad}$-5), on three aliquots, either in one measurement sequence (most tests) or in three individual sequences ($T_{fad}$-15). Whereas in some cases, the measurements on more than one aliquot showed identical results within the

expected scatter of the data values thus representing merely repeated measurements, in other cases the position of an aliquot within a measurement sequence mattered. To demonstrate this issue, we decided to show the results of all three aliquots for each fading test presented in sections 3.2 and 3.3. An overview on the tests presented in sections 3.1 to 3.3 is given in Fig. 1.

*figure 1 near here*

---

[1] First illustrated means that for the fresh aliquots the first SAR cycle with the natural luminescence signal (N or N+ß) was not considered for graphical display. Likewise, for used aliquots on which at the beginning of a test five SAR-cycles with zero delay (prompt IRSL readout after laboratory irradiation and preheating) were routinely measured, the first two precursor cycles were neglected and only the last three zero-delay cycles were illustrated together with the following cycles with longer pauses. These details, however, are owed to data sheet templates, but are not relevant for the interpretation of the results. Details on this issue may be tracked in supplement 2, columns BM – BP.



### 3.1 Tests on reader DA12 and changeover to DA20 on loess-borne samples from SW-Germany with preheat 120 s at 220 °C – IRSL readout without *vs*. IRSL readout with thermal assistance

#### 3.1.1 Tests on reader DA12 – Testing IRSL at room-temperature *vs*. IRSL plus additional thermal input

Fading test 1 ($T_{fad}$-1) was carried out in 2012 on a loess-borne sample (HDS-504) from southwestern Germany (Fig. 2). When measured with fading tests in the course of an MAA protocol in earlier studies (e.g., Kadereit et al. 2010; Lang & Wagner 1996; for details cf. supplement 1), the IRSL signal (blue-violet, 410 nm)[2] of polymineral fine-grains of loess-borne samples from SW-Germany had proved stable. However, when the SAR-protocol with the MAA-like preheat procedure (120 s at 220 °C) and the MAA-like IRSL at room-temperature was applied in the present study using the SAR protocol (on the same reader DA12 as used earlier for the MAA measurements), strong signal loss was observed (Fig. 2a). If that trend of signal loss observed in the laboratory is tentatively graphically extrapolated to geologic time scales, after a few $10^5$ years a significant part of the signal would be gone (Fig. 2b). This result is in strong contrast to the findings of Lang et al. (2003), who for the blue-violet IRSL signal could show agreement of IRSL ages with independent age control up to ~ 120 ka. When applying the same $T_{fad}$-1-like measurement to samples from different areas, similar results were found (not illustrated here). Strangely, the signal loss does not follow a logarithmic decay function. Rather, the values seem to form a plateau for pauses ≤ 20 s and show a strong decline for pauses of 20 – 60 s as well as a lesser decline for pauses > 60 s. This specific form of IRSL decline could be observed only because the number of data points were unusually high and the pause intervals were unusually narrow for fading tests. We tentatively omitted the results of the shorter pauses and every second result of the longer pauses in Fig. 2c to mimic a more common fading test with fewer measurement points. In awareness of the course of the complete set of data points, it would obviously be inappropriate to describe the decline of the remaining values in Fig. 2c with a linear decay function (cf. "erroneous interpretations"), as conventionally done for *g*-value assessment.

*figure 2 near here*

In a further test on the same aliquot ($T_{fad}$-2) the range of pauses from 0 s to 120 s (covering the initial plateau, the strong decline, and the transition to a gentler decline as observed in $T_{fad}$-1, Fig. 2) was investigated (Fig. 3a). With the findings of Rhodius et al. (2015; cf. section 1.5) in mind, the non-logarithmic signal decay might suggest that the results for the shorter pauses represent IRSL-signals that are more temperature-assisted than those of the longer pauses, which are inevitably accompanied by increasingly longer time periods for the heating plate to cool down in between preheat and IRSL readout. Therefore, two further subtests were included for the declining range 30–120 s with (1) IRSL after a warming (cutheat, i.e., ramp similar to a preheat to the required temperature but not held for several seconds or minutes) of the aliquot to 60 °C and (2) IRSL-readout at 60 °C, with 60 °C being the liftup temperature in the respective measurement sequence. While IRSL-readout at 60 °C produced normalised SAR-corrected values well above 1 (up-pointing triangles in Fig. 3a), the values from the warmed-up aliquot scattered around unity (diamonds). As assumed by Rhodius et al. (2015), the control of the IRSL-readout temperature appears highly relevant to fading tests.

As, however, the brief cutheat to 60 °C prior to IRSL-readout at room-temperature in $T_{fad}$-2 (Fig. 3a) might be interpreted to stimulate electron redistribution (*sensu* Auclair et al. 2003), we repeated and extended that same test on a different loess-borne sample from SW-Germany (HDS-713) on reader DA12 in 2016 ($T_{fad}$-3, Fig. 3b). $T_{fad}$-3 showed results comparable to those of $T_{fad}$-2, despite the liftup temperature this time being 20 °C, which compared to the room-temperature in the luminescence laboratory at the time of measurement. Additionally, for pauses ≥ 30 s we included a further subtest, with heating not on the sample position but on a different (and empty) turntable position after the pause – thus inevitably elongating each pause prior to IRSL-readout for 120 s, which should increase the total amount of fading according to the law of logarithmic decay. Instead,

---

[2] detection through a set of glass filters BG39, 2 x BG3, GG400 (Schott, 3 mm each) following Krbetschek et al. (1996)



the values (cf. cross symbols in Fig. 3b) continue the plateau of the pauses ≲ 20 s, now up to 120 s. Therefore, this subtest demonstrates that electron charge redistribution (*sensu* Auclair et al. 2003) cannot be responsible for the observed phenomenon but that a sufficiently or insufficiently controlled IRSL-readout temperature causes either the consistency (continuing thermal assistance of IR-stimulated eviction and recombination of electrons) or the decline (waning thermal assistance of IR-stimulated eviction and recombination of electrons with increasing delay time) of the SAR-corrected values. Although the IRSL-signal ($T_x$) of the repeatedly administered normalisation dose (NRM) may correct for multifold changes of the boundary conditions during the preceding measurement of the signal ($L_x$) of the corresponding laboratory dose (LAB), it may not correct for varying readout-temperatures of LAB as IRSL-readout of NRM occurs always promptly after the preheat and not under identical conditions as for LAB. This is in contrast to the MAA-fading test (cf. supplement 1), in which an extra set of aliquots with the natural signal ($N_{fad}$) would also correct for possibly varying readout-temperatures between the actual dose measurements and the fading measurements. Therefore, it is plausible that samples may show fading with the SAR protocol and no fading with the MAA protocol.

$T_{fad}$-1 to $T_{fad}$-3 were run in an $N_2$-saving mode, i.e., nitrogen flow occurred only for 120 s (quasi-manually induced by using the Risø software Test500 for the manual operation of TL DA12) immediately before the start of the actual measurement which then was run in the $N_2$ atmosphere (no previous generation of vacuum). When $T_{fad}$-3 was repeated on another fresh aliquot of sample HDS-713 with continuous $N_2$ flow (results not shown here), the different $N_2$ mode did not reveal any noticeable impact on the shape of the data curve.

*figure 3 near here*

### 3.1.2 Reader DA20 - Testing IRSL at room-temperature *vs.* IRSL plus additional thermal input

$T_{fad}$-3 was repeated in Dezember 2016 on reader Athenaeum with a liftup temperature of 24 °C and continuous $N_2$ flow ($T_{fad}$-4, Fig. 3c) as well as without $N_2$ flow (not shown here). The results of both tests appeared identical. However, unlike the measurements on reader DA12 all subtests of $T_{fad}$-4 with IRSL-readout at room-temperature produced double normalised SAR values close to unity. This indicates that the temperature control of luminescence readers has been significantly improved. The double normalised SAR-values for IRSL-readout at 60 °C are larger on reader Athenaeum (in the range of 1.3 to 1.4 instead of around 1.2 on DA12 which could point to stronger heat-assistance or heat-accumulation in the newer reader), but they show a conspicuous decline of ~ 5 % in the range of pauses from 30 s to 120 s. This would suggest respective fading at elevated IRSL-temperature which is not observed for IRSL at room-temperature. Yet such a decline was not observed on reader DA12, and the shape of the decline does not conform to the model of logarithmic decay.

### 3.2 Tests on reader DA20 on dim samples from Lake Chapala/Mexico with preheat 60 s at 250 °C and IRSL at elevated temperatures (50 °C, 60 °C) and at room-temperature

Due to the dimmness of polymineral fine-grains of limnic samples collected near Satillo on Lake Chapala in Mexico, fading tests in the frame of MAA measurements had not produced unequivocally interpretable results (Kadereit et al. 2017). Therefore, these samples were subjected to additional fading tests using the SAR protocol. As the MAA protocol included a preheat of 60 s at 250 °C and IRSL-readout at room-temperature, these same parameters were used for the SAR approach (section 3.2.2). For better comparison with the study of Auclair et al. (2003), in addition elevated temperatures of 50 °C and 60 °C were applied for IRSL-readout (sections 3.2.1, 3.2.3). As the samples proved quite dim, despite rather large LAB and NRM of ~ 41 Gy (cf. section 2.3), we also included some tests on the brighter loess-borne samples from SW-Germany for comparison (cf. section 3.3). All these measurements were carried out on the reader Athenaeum (DA20) with a ~ 3.7 times stronger β-source than reader DA12, which reduces measurement times accordingly. Basically, two types of fading protocols were applied, (1) one which covers pauses only up to 14,400 s–36,000 s (4–10 h) but includes a higher number of short pauses



(cf. Fig. 4) and (2) one which covers pauses up to 64,800 s (18 h) but includes only a thinned-out number of measurement points (cf. Fig. 6a-c). The latter follows Auclair et al. (2003), but includes one semi-short pause of 196 s. In addition, it repeats

the prompt IRSL-readout (denoted as "0 s delay" on the x-axes of Fig. 4 ff.) several times both at the beginning and at the end of each fading test and also repeats the 196 s delay towards the end of the test. Such repetitions are considered as relevant, as beyond the LAB-by-LAB sensitivity changes monitored by the usual SAR-correction, they may monitor trends of sensitivity changes throughout a complete test measurement. Further, these repeat measurements provide an idea of the possible scatter of the values and therefore a more solidly grounded interpretation of the values connected to the longer pauses. In cases when

more than one aliquot was measured, longer pauses ≥ 6,000 s (1.7 h) were taken together for all aliquots to save measurement time. This means that the length of pauses was *de facto* slightly longer for the aliquots measured subsequently to aliquot #1, as for these IRSL-readout time of the previously measured aliquots plus, depending on the type of fading test, the time of preheating would add up to the actual pause. However, as these differences are minor and as we do not intend to calculate true *g*-values, this issue was not considered in the graphical display of the results of the tests. Instead, it was pretended that not only

the shorter pauses but also the longer pauses were identically long for all aliquots measured in one sequence. Yet this simplification does not affect the overall shape of the data curves.

### 3.2.1    IRSL 60 °C and liftup temperature 60 °C – Testing one aliquot *vs*. a sequence of three aliquots

T$_{fad}$-5 and T$_{fad}$-6 were performed on three previously used aliquots of the limnic sample HDS-1712 (Lake Chapala) distributed on turntable positions 1, 7 and 13 on reader Athenaeum to avoid possible effects of potential cross-bleaching (cf. Kreutzer et

al. 2013). IRSL-readout occurred at 60 °C, after 10 s warmup. The liftup temperature was 60 °C. In T$_{fad}$-5 and T$_{fad}$-6 the pause was placed in between preheating and IRSL-readout as recommended by Auclair et al. (2003). In T$_{fad}$-5 (Fig. 4a) only one aliquot was measured (position 7) whereas in T$_{fad}$-6 (Fig. 4b-d) three aliquots were measured in one sequence. While the one aliquot in T$_{fad}$-5 does not show any sign of signal fading (Fig. 4a), the three aliquots of T$_{fad}$-6 mostly show some trend of signal decline (Fig. 4b+d). However, whereas for position 1 the final prompt readouts (0 s, after the break in the x-axis in Fig. 4b)

show by trend higher values than those for the longer pauses pointing to signal loss, the final prompt readouts for the aliquot on position 13 continue the downward trend of the longer pauses (Fig. 4d). With regard to T$_{fad}$-5 some sort of heat accumulation in the immediate environment of the aliquot could explain these observations best. Again, the course of the data values is not compatible with logarithmic decay, as the declining trend is observed mostly for pauses ≳ 60 s, which, too, supports the assumption of some kind of varying heat accumulation, assisting the IRSL-readout sometimes more (short pauses) and

sometimes less (longer pauses). Although the effects are not as clearly visible as for IRSL-readout at room-temperature on the reader DA12, they would significantly affect potential *g*-value calculations towards an overestimation. True anomalous fading should be detectable in equal measure irrespective of the number of aliquots measured in one sequence. The measurements show also that it is important to incorporate more than one immediately measured (zero delay) dose point. Possible fading cannot be evaluated against only one zero-delay dose point but has to account for the full range of the data scatter. In dating

measurements ± 10 % deviation of a repeatedly measured dose point is usually regarded as an acceptable recycling ratio, and should therefore be expected also in fading tests. Further, the results, e.g., for the aliquot on position 13 (Fig. 4d) show the importance to measure dose points with zero delay also at the end of a sequence. Not only the range of the data scatter of dose points with zero delay at the beginning of a sequence may determine whether or not a possibly downward trend of the data from dose points with increasing delay time has to be assessed as signal loss, but also whether the complete sequence shows a

downward trend of the data points which is not fully corrected by the SAR protocol.

*figure 4 near here*





### 3.2.2 IRSL at room-temperature – Testing liftup temperature of 60 °C *vs*. 24 °C

Further tests on the same Lake Chapala aliquots were performed with IRSL readout at room temperature with the liftup
temperature at first being 60 °C. In accordance with the observations of Rhodius et al. (2015) and Auclair et al. (2003) no
fading was observed when the tests were performed with the pause before the preheat ($T_{fad}$-7, Fig. 5a-c; cf. green line as a
guide for the eye) as suggested by Rhodius et al. (2015) and is present if the pause is inserted after the preheat ($T_{fad}$-8, Fig. 5d-
f; cf. pink line) as suggested by Auclair et al. (2003). In comparison with IRSL-readout at 60 °C ($T_{fad}$-6, Fig. 4b-d) the signal
loss for IRSL-readout at room-temperature ($T_{fad}$-8, Fig. 5d-f) turns out being more dramatic, especially for the later measured
aliquots. The lesser decline for the first aliquots is explained by less intense heat accumulation for the first aliquot in a row of
three successively measured aliquots. Again, a kind of plateau is observed for the shorter pauses, here ≲ 40 s, followed by a
stronger decline up to ~ 7,200 s (2 hours), before the declining trend seems to fade out. This observation is in contradiction to
a fading mechanism following logarithmic decline. However, any indication of seeming fading could be eliminated, when in
$T_{fad}$-9 (Fig. 5g-i) for pauses ≥ 120 s neighbouring empty turntable positions (here, e.g., 46–48 for the aliquot on position 1)
were heated (here each position for 180 s at 250 °C, which compares to the preheat temperature on the measuring positions)
after the actual pauses, thereby actually extending the pauses (cf. $T_{fad}$-3). Although, according to the law of logarithmic decay,
such longer delay times should reduce the remaining luminescence signal, like for the respective subtests on the loess sample
from SW-Germany on reader DA12 in $T_{fad}$-3, the samples from Lake Chapala on reader Athenaeum did not show any increased
fading, but no indication of fading at all. This corroborates the assumption that some sort of heat accumulation assisting the
400 IR-stimulated electron eviction is responsible for the phenomena observed in $T_{fad}$-7, $T_{fad}$-8 and $T_{fad}$-9, rather than true
anomalous signal fading (Wintle 1973) or charge redistribution (Auclair et al. 2003). Any indication of signal fading could
also be eliminated if for IRSL readout at room temperature the liftup temperature was reduced to 24 °C (cf. $T_{fad}$-10, Fig. 5j-l).
This shows that for fading tests the liftup temperature needs to be adjusted appropriately low for a given readout temperature,
even though this may elongate the time to complete a measurement. At present, the liftup temperature may be preset only for
a complete measurement sequence uniformly and may not be adjusted for individual operations. Thus, a liftup temperature of
24 °C, which is appropriate for IRSL readout at room temperature, will require unnecessarily long cool-down times, e.g., prior
to preheating at 250 °C.

*figure 5 near here*

### 3.2.3 IRSL at 50 °C – Testing dense data points *vs*. thinned out data points

In Fig. 6 the results are compared for two fading tests on Lake Chapala sample HDS-1712 for IRSL readout at 50 °C and a
liftup temperature of 60 °C. In both cases extra heating was performed on neighbouring turntable positions after the longer
pauses (3 x 60 s at 250 °C, which compares to the preheat temperature), which, in view of $T_{fad}$-9, should eliminate possible
signal decline (cf. $T_{fad}$-9, Fig. 5g-i). The two tests differ primarily in the number of measured dose points, plus that in the test
with the thinned-out number of dose points not only the zero-delay dose point but also the 196 s dose point (shortest pause)
was repeated at the end of the measurement sequence (cf. symbols after the break of the x-axes in Fig. 6a-c, $T_{fad}$-11). The
measurement with the dense data points clearly shows no signal decline (Fig. 6d-f, $T_{fad}$-12). In contrast, the measurement with
the thinned-out number of LABs appears less straightforward to interpret. Strangely enough, a decline of data values, as
observed for the aliquots on position 7 and 13, starts only for longer pauses (> 196 s), which does not conform to logarithmic
decay, but may be owed to some heat accumulation for the first measured dose points (zero or short delay) as compared to the
420 later measured dose points. This shows that an increased number of dose points, including the repeated prompt IRSL readout
of LAB, are beneficial for the interpretation of the data.

*figure 6 near here*





### 3.3 Tests on reader DA20 on a bright loess-borne sample from SW-Germany with preheat 60 s at 250 °C, 20 s at 280 °C and 60 s at 280 °C and IRSL at elevated temperatures (50 °C, 60 °C)

Different samples from different areas, as those from Lake Chapala in Mexico and from SW-Germany, may exhibit different fading characteristics. However, as the loess-borne samples from Germany are brighter, they might offer the opportunity to reveal characteristics which may be camouflaged by the scatter of the values of the dimmer samples from Mexico. Therefore, in the following, results of 6 tests on the loess borne sample HDS-713 are presented. The measurements were carried out on turntable positions 21, 27 and 33 of reader Athenaeum (DA20). The tests were performed with a laboratory dose of 100 s

(9.6 Gy) and a normalisation dose of 50 s (4.8 Gy). Preheat was either 60 s at 250 °C ($T_{fad}$-13 – $T_{fad}$-16, Fig. 7a-l), 20 s at 280 °C ($T_{fad}$-17, Fig. 8a-c) or 60 s at 280 °C ($T_{fad}$-18, Fig. 8d-f). IRSL readout temperature was set to 50 °C when the preheat temperature was 250 °C and to 60 °C when preheat occurred at 280 °C. The liftup-temperature corresponded in each case to the IRSL-readout temperature. All tests – with the exception of one test for comparison ($T_{fad}$-16, Fig. 7j-l) – were measured in the way as recommended by Auclair et al. (2003), i.e., with the pause immediately after irradiation and preheating.

As an overall result, none of the tests produced data corresponding to logarithmic signal decay. All tests showed a kind of plateau for the shorter delay times and a decline of data points for the longer delay times. If the part of the initial semi-plateau is regarded to represent above-average temperature-assisted IRSL-readout, then the declining part of the data curve (up to ~ 10,000 s or 2.8 hours, respectively) may be regarded as IRSL-readout with decreasing temperature assistance rather than true anomalous fading. However, the shape of the data curve, especially the length of the initial plateau, varied with the variation

of additional measurement parameters.

#### 3.3.1 Preheat 60 s at 250 °C, IRSL 50 °C and liftup-temperature 50 °C – Testing initial N₂ flooding *vs*. continuous N₂ flow, one aliquot *vs*. three aliquots per sequence, extra heating on neighbouring turntable positions, liftup-temperature of 60 °C *vs*. 24 °C, pause after preheating *vs*. pause prior to preheating

     The shortest plateau, especially for the first two measured aliquots on positions 21 and 27, was observed for $T_{fad}$-13 (Fig. 7a-

c), in which N₂ flooding occurred only immediately prior to the beginning of the fading test. The initial plateau covers pauses up to ~ 30–120 s, before values drop up to ~ 5 % below the values representing zero delay. The values for the longest pauses (> 10,800 s, 3 h) might represent data scattering at a lower level rather than continuous decline following the law of logarithmic decay. Therefore, the data curve may be tripartite (initial plateau, steep decline, lower plateau) and the data values may bottom out beyond a certain minimum length of the pauses. Such behaviour would be compatible with observations by Steffen et al.

(2010) and Trauerstein et al. (2014, Fig. 6a) who also found an expiring of the fading after a while.

*figure 7 near here*

     $T_{fad}$-14 compares to $T_{fad}$-13, but in contrast to the latter fading test it was performed with continuous N₂ flow, instead of only 120 s N₂ purge at the start of the SAR measurement. Compared to the fading test without continuous N₂ flow the intitial part of the (semi-)plateau of the double normalised SAR values (30–120 s for $T_{fad}$-13) is extended up to pauses of 600 s (10 min)

(Fig. 7d-f). Like in the previous test ($T_{fad}$-13) the values representing the longest pauses seem to scatter at a lower level, rather than to continue a decline as expected for true anomalous fading.

     In a further test ($T_{fad}$-15, Fig. 7g-i) the aliquots were analysed separately in three individual SAR runs. This variant is similar to a SAR protocol version with aliquots run one at a time (but, of course, without the possible heat influence on an aliquot from previously measured aliquots). This, too, leads to an elongated initial (semi-)plateau of the double normalised SAR values

up to pauses of 1800 s, 3600 s or even 7200 s (0.5–2 h), depending on where one tends to divide the data curve by mere visual examination. Irrespective of the tentatively placed points of separation, elongation of the initial (semi-)plateau is more efficient than with continuous N₂ flow, but the values drop subsequently sharply to a level ~ 4–6 % below the level of the data points





representing zero delay. Scattering at the lower level, as observable in the previous tests $T_{fad}$-13 and $T_{fad}$-14 is not discernable in $T_{fad}$-15.

Compared to $T_{fad}$-13 (Fig. 7a-c) the initial (semi-)plateau may also be elongated by extra heating on neighbouring turntable positions, by lowering the liftup temperature to 24 °C (both tests not shown here) or if the pause is placed before the preheat ($T_{fad}$-16, Fig. 7j-l)), as recommended by Rhodius et al. (2015). Although this mode provides readout temperatures as homogeneous as possible throughout a complete SAR fading test, it nonetheless leads to a data curve not compatible with simple logarithmic signal decline. It seems that during the measurement period of the dose points with shorter pauses (here ~

600 – 2400 s, i.e., up to 40 min), likely due to the repeated preheating and IRSL readout at elevated temperature in short intervals, some kind of heat accumulation assists the IRSL readout in the luminescence reader. That the IRSL signal increases with increasing readout-temperature has been shown earlier, e.g., by Habermann (2000) and was confirmed in the course of the present study for Lake Chapala sample HDS-1712, which in the temperature range 41–80 °C showed a signal increase of ~ 1 % per additional K readout-temperature (cf. supplement 1). In a reverse conclusion, ~ 4–5 % signal decline, as observed

in most of the here presented fading tests, could correspond to a decrease of the readout temperature of ~ 4–5 °C, or the other way around, a ~ 4–5 °C higher temperature for the shorter pauses than for the longest pauses. However, such a conclusion would be correct only, if all the observed signal decline was caused by a difference in readout temperature. That readout temperature plays a role, could be shown by the tests including additional heating on neighbouring turntable positions (e.g. $T_{fad}$-9, Fig. 5g-I, section 3.2.2). Nevertheless, some of the decay may be caused by true anomalous signal fading as originally

defined by Wintle (1973). However, if both (1) true anomalous fading and (2) variations in the temperature assistance of the IRSL signal are responsible for the shape of the data curve, the proportion of true anomalous fading cannot be deduced from the SAR corrected data points. These would only give a possible maximum range of the non-thermally assisted fading of an IRSL signal. The best precaution to provide readout temperatures as homogeneous as possible for each data point is to preheat an aliquot immediately prior to IRSL readout. Such a procedure should narrow down the maximum degree of fading, which

may be derived from SAR fading tests, as much as possible.

### 3.3.2 Preheat 20 s at 280 °C and 60 s at 280 °C, IRSL 60 °C and liftup-temperature 60 °C – Testing shorter *vs.* longer preheat time and a later *vs.* an earlier position of an aliquot in a sequence

$T_{fad}$-17 (Fig. 8a-c) and $T_{fad}$-18 (Fig. 8d-f) on sample HDS-713 were performed on reader Athenaeum (DA20) on turntable positions 40, 44 and 48, respectively. IRSL-readout temperature and liftup-temperature were 60 °C. For these two tests,

however, the preheat temperature was 280 °C. The tests were performed with continuous $N_2$ flow. Whereas the preheat duration of $T_{fad}$-17 was 20 s, it was elongated to 60 s in $T_{fad}$-18. $T_{fad}$-17 exhibits very short initial plateaus, i.e., for pauses up to ~ 20 s, before values drop to a lower level, around which they scatter for the longest pauses. The three times longer preheating at a relatively high preheat temperature of 280 °C in $T_{fad}$-18 leads to much longer initial (semi-)plateaus, which include pauses up to 14,400 – 18,000 s (~ 4–5 h) (Fig. 8d-f). The initial part of the data curve of the third aliquot (position 48;

Fig. 8f) measured in one row seems to overshoot the initial plateau level for the shorter pauses ≲ 600 s (10 min), likely as a result of increased heat accumulation due to repeated and long heating in short intervals. The difference between the largest (~ 1.04) and the smallest values (~ 0.94) amounts to 10 %. With regard to pure anomalous fading, it would not be expected that aliquots measured after stronger preheating (here 60 s at 280 °C) exhibit up to two times stronger signal decline than aliquots measured after more moderate preheating (here 60 s at 250 °C as in $T_{fad}$-13, $T_{fad}$-14, $T_{fad}$-15). A larger difference in

IRSL readout temperatures between the shorter and the longer pauses for the fading test with the stronger preheat procedure may explain that phenomenon. Thus, $T_{fad}$-17 and $T_{fad}$-18 (Fig. 8) show again that the data curves of the fading tests do not follow the expected course of logarithmic signal decay. Moreover, the shape of the data curves may be manipulated by varying the SAR parameters (here the length of the preheat duration and the position of an aliquot in a sequence of three measured aliquots).



*figure 8 near here*

## 4 Discussion and conclusions

### 4.1 Hypotheses on the origin of the initial plateau of $L_x/T_x$ fading data

In contrast to ordinary SAR based fading tests as usually performed within dating studies, the tests compiled in the present study include much more data points, i.e., especially such with very short to short delay times (seconds to minutes and few

510 hours). This variation of the SAR fading test protocol reveals that the course of the SAR values may not follow the law of logarithmic decay, as would be expected for "true" anomalous fading *sensu* Wintle (1973) which results from quantum-mechanical tunneling and localized transitions (e.g., Visocekas 1985, Templer 1986). Instead, the course of the values may exhibit an initial plateau for the shortest pauses, before the values start to decline for the longer pauses. Depending on the SAR protocol parameters, the declining course may be subdivided, starting with a section of retarded decline of the data points, and

515 for longer pauses (several hours) often ending in a section, which does not seem to follow steady logarithmic decline either, but which may as well represent scattering around a lower level. The absence of an initial decline of the SAR values may be interpreted in different ways.

(1) Electron redistribution might occur after laboratory irradiation and/or after preheating (cf. Molodkov et al. 2007; Auclair et al. 2003). Such processes could possibly interfere with, or rather superimpose, processes like tunnelling and localized

transitions which are regarded to cause the fading of the IRSL signal. In both of these cases electrons originally captured in the IRSL trap would have left the trap at the time of measurement so that they could no longer contribute to the IRSL signal by radiative recombination at the sampled IRSL center during IRSL readout. In the case of an initial (semi-)plateau, however, some latent additional IRSL signal must countervail the supposedly fading IRSL signal. Assuming that preheating causes this excess of the IRSL signal the additional electrons might possibly be provided by thermal release from IR-insensitive traps into

the IRSL trap so that they would be sampled during the subsequent SAR IRSL-readout. As, however, the initial (semi-)plateau of the fading curves turns into a downward trend after a while (seconds e.g. in $T_{fad}$-17 to hours e.g. in $T_{fad}$-15), this part of the IRSL-signal would have to be regarded as highly instable or short-lived. It could therefore represent an additional, but only short-term fading component, or it might be caused by ordinary thermally assisted electron escape. In both cases this component would not be relevant for long term fading of a sample over geologic time scales. Therefore, it must not be included

in the determination of *g*-values which are used to calculate fading-corrected IRSL ages by adjusting the potential IRSL signal loss which might have occurred over the whole dating period. As, e.g., stronger and longer preheating might lead to a stronger and longer contribution of the unwanted short-term component (cf. $T_{fad}$-18 *versus* $T_{fad}$-17), one would need to observe a sample's behaviour under a specific SAR protocol, to decide after which lag time it is possible to determine only the long-term fading component relevant for the IRSL dating. Fading tests, as compiled in the present study, which start with very short

pauses and elongate the pauses gradually, could provide a procedure for monitoring the samples from which time onwards the fading signal follows the law of logarithic decay. If, however, the short pauses after laboratory irradiation may not be included in the fading tests, this would vitiate the alleged advantage of the SAR protocol for *g*-value determination, i.e., to monitor several decades in a comparably short time span in the luminescence reader in the laboratory (cf. Auclair et al. 2003). It should be noticed that electron redistribution was not proved in the present study and therefore has to be regarded as a hypothetical

explanation for the observed shape of the fading curve.

(2) Visocekas (1985) reinvestigated fading of the Wintle (1977) labradorite sample in liquid $N_2$, i.e. at temperatures ≤ 77 K (- 196 °C). This particular experimental setup allowed observing tunneling afterglow after the irradiation of the sample, which, depending on the length of the laboratory irradiation, may last from seconds to several hours. In accordance with observations from Delbecq et al. (1974) and Molodkov et al. (2007), mere tunneling-related fading following logarithimic decay could be

observed after a period corresponding ~ 10 times the duration of the laboratory irradiation. These observations may suggest





that for fading tests a minimum lag time has to elapse before the luminescence signal may be read out. Again, like for (1), fading tests with gradually increasing pauses may be a means to monitor from which time onwards the fading signal follows logarithmic decay. If this model is accepted, a promptly read out dose point, like the shorter pauses influenced by tunnelling afterglow, must not be included in the $g$-value calculation. Considering our experimental data and the onset of the decline of

the $L_x/T_x$ values in the SAR fading tests at delay times in the range of mostly a few 100 s, it appears that this explanation cannot fully account for the observations. If "pure" tunneling recombination would dominate, an even longer initial plateau of $L_x/T_x$ values would be expected (in the order of 1,000–4,000 s) given the duration of laboratory regenerative irradiation (100–400 s). It is, however, conceivable that the effect of pure tunneling is supplemented by other mechanisms, which somewhat impede or blur the formation of an initial plateau as related to mere tunneling recombination. One such mechanism could be the

declining heat assistance in a luminescence reader with increasing delay time (cf. explanation 3).

(3) In addition to the two theories mentioned under (1) and (2) to explain the initial plateau of the fading curves, some of the tests of the present study indicated that the IRSL readout temperature influences the course of the fading curves, i.e., especially the initial part corresponding to the shorter pauses. Such obvious influence of the temperature was demonstrated by the differing results of measurements with the liftup temperature, on the one hand, comparing to the IRSL readout temperature

and, on the other hand, well above the IRSL readout temperature. Further, the (initial) decline of the fading curves could be "detrended" by heating prior to IRSL readout, not at the sample position but at neighbouring empty turntable positions. Whereas these procedures (or rather SAR parameter configurations) worked well for IRSL readout at room-temperature, they could not reduce the signal decline comparably for IRSL readout at elevated temperature. Nonetheless, the SAR measurements with IRSL readout at room-temperature and a too high liftup temperature or extra heating on the non-measurement positions

prior to IRSL readout on the measurement positions demonstrated that the shape of the fading curves may be manipulated by heat input. As could further be corroborated in the present study, 1 K increase of the IRSL readout temperature leads to ~ 1 % increase of the SAR corrected IRSL signal strength. In contrast to MAA fading tests, which may correct also for potentially different readout temperatures at the different dates of delayed fading measurements, such a correction may not be provided by the SAR protocol. Therefore, it is hypothesized that (slightly) varying temperatures during IRSL readout have an

(additional) influence on the shape of the fading curves. This hypothesis is corroborated by findings of Guérin and Visocekas (2015) which imply that the temperature very much controls the thermal fading (i.e., localized transitions, hopping along band tail states) happening in a feldspar sample. A consequence would be that a valid fading rate could be determined only if the temperature experienced by a sample during burial in nature was known and could be reproduced in the laboratory. Further support for this hypothesis is lent by the general feldspar luminescence model established for instance by Jain and Ankjærgaard

(2011), in which the long-term signal stability is inherently coupled to the thermal energy supplied prior to and during measurement. Hence, not only IRSL signal intensity but also long-term stability appears to be closely related to the thermal treatment of the feldspar sample, which is elaborated in more detail in the following in view of the experimental findings of this study.

### 4.2    Influence of the thermal regime of samples on fading measurements

The observed decline of data values on the newer luminescence reader (DA20) is in the range of ~ 5–10 % in the tested range of delay times, which would correspond to a difference in the readout temperature of $\Delta$ 5–10 K. As this magnitude appears to be quite large, probably not all of the decline of the data values may be explained by varying heat assistance of the IRSL signal. An undeterminable portion of the decline may be caused by true anomalous fading *sensu* Wintle (1973). Also, part of the initial (semi-)plateau or retarded decline of the data values may possibly be explained by the processes explicated under (1) and/or

(2), yet likely overprinted by heterogeneous readout temperatures. The non-thermally induced signal fading ("true" fading), however, may at present not be subtracted from the total decline. Therefore, if SAR fading tests are performed, they may only give a maximum estimate of the actual fading of a sample and are therefore not suitable for calculating $g$-values which are



used to correct IRSL ages for longterm fading. Signal decline on the older luminescence reader (DA12), which was run with an out-of-date steering software, is even larger than on the modern reader. Plainly obvious, those fading tests produced erroneous results which, if taken seriously, would mean that loess-borne samples from southwestern Germany would loose much of their IRSL signal within a few $10^4$–$10^5$ years, which is in strong contrast to earlier studies (e g., Lang et al. 2003). In the present case it is most likely that the emulation software did not transfer the command from the sequence editor that the sample should be treated further only if the temperature has fallen below a predefined value (liftup temperature). The ongoing hard- and software development has led to much better temperature control of modern readers, but SAR-based fading tests of older studies should be regarded sceptically, especially if $g$-values are unusually large.

Measurement parameters for fading tests based on the SAR protocol – like for the actual SAR-dating measurements itself – are manifold. Usually, only a few parameters are detailed in publications, such as, e.g., the duration and temperature applied for preheating. However, the series of tests compiled in the present study shows that other parameters may influence the heat accumulation in the immediate environment of an aliquot. This affects the readout temperature during IR-stimulation and therefore the results of a SAR fading test, and finally the size of a resulting $g$-value. Among the factors of influence are the number of aliquots measured in one sequence, the use and the mode of $N_2$ flow and the software version controlling the heating. If a dating measurement of other aliquots is intercalated between the measurements of SAR fading dose points with longer delay times, in order to make good use of the idle time in the reader, the shape of the data curve of the SAR fading dose points may be influenced by the measurement of these other aliquots. This scenario was simulated in the present study by applying extra heating on neighbouring empty turntable positions. Further likely paramaters influencing the results of a fading test are the form, material and thickness of the carriers on which the analysed mineral grains are placed, which were not varied in the present study. However, different $g$-values were produced for feldspar separates once measured as single aliquots and once as single grains, i.e. on different sample carriers (cf. Trauerstein et al. 2014). Also, the protocol variant of measuring one aliquot at a time, which here was not further investigated either but only approximated by measuring the three aliquots of a fading test in three different measurement sequences, would influence the actual readout-temperature and therefore the size of a $g$-value. In short, the parameters of a SAR protocol are too manifold to be globally standardised for inter-laboratory comparison of $g$-values.

Basically, each measurement step preceding IRSL-readout influences the temperature-assistance associated with the IRSL-readout, especially if the preceding measurement steps, on the actual aliquot or on another position in the reader, include long and strong heating. Thus, each IRSL-readout is dependent on previous measurement steps in a measurement sequence.

Our tests show that preheating immediately after laboratory irradiation cannot be recommended, as this way constant temperatures for IRSL readout after varying delay times may least be expected, even on modern fully-automated luminescence readers with recently updated software developed for more effective temperature control of the heating unit. If readout temperatures are highest for immediate IRSL readout (supported by heat accumulation originating from the immediately preceding preheating) and decline with increasing delay times (as accumulated heat gradually dissipates), $g$-values will by trend be overestimated. Assuming that the measured data points represent the true anomalous fading of a sample and thererfore follow a linear trend when represented in a semi-logarithmic graph, it appears to have become an unfortunate practice to measure only very few data points (e.g., one immediately or shortly after the end of the laboratory irradiation and two further ones after one and two days delay time, respectively), which then are fitted with a model of logarithmic decay to calculate a respective $g$-value. This way, however, the more complicated nature determining the actual course of the data points will stay unnoticed, and too high $g$-values will be generated as a consequence of applying an inappropriate fitting model.

Auclair et al. (2003) argued that unwanted electron redistribution after irradiation may be responsible for lower $g$-values if preheating is not performed immediately after irradiation (pause between preheat and IRSL readout) but prior to IRSL-readout



(pause between irradiation and preheat). However, experiments with extra heating on neighbouring turntable positions carried out in the present study showed that electron redistribution cannot be the (only) cause of $g$-value underestimation. Further, during SAR dating measurements preheating is generally performed immediately prior to IRSL readout (usually no extra pauses are intercalated in between measurement steps as this would elongate the total measurement time), and preheating prior to IRSL-readout of the natural dose occurs after a long storage in nature. Applying a measurement procedure with such highly variable times of preheating with respect to the time of irradiation implicitly assumes that the timing of the preheating must 635 be irrelevant, as otherwise $D_e$ values determined with a SAR protocol could not be used for age calculation.

### 4.3    Practical considerations

The unwanted effects of heat accumulation and conduction are strongest for IRSL readout at room-temperature and if the liftup-temperature is not well adjusted. The respective examples compiled in the present study illustrate the potential problems arising from insufficient temperature control in the frame of SAR-based fading tests best. Nonetheless the unwanted effects
affect also classical IRSL readout at elevated temperature (here: 50 °C, 60 °C). It may further be hypothesized that fading tests in the framework of pIRIR dating may be affected as well. Although, on the one hand, pIRIRSL readout occurs at much higher temperatures (usually in the range 150–300 °C), which would reduce the malign effects, on the other hand, preheating, which mainly causes the effects, takes much more room in pIRIR protocols. This assumption is in accordance with Thiel et al. (2011) who considered pIRIR $g$-values from samples which had produced the expected ages reliably as likely measurement artifacts.
It may further be expected, that IRSL $g$-values determined in the course of pIRIR measurements (e.g., $pIR_{50}IR_{225}$) are strongly affected by heat accumulation and conduction, as heating is omnipresent in the pIRIR protocols and their derivates (prolonged and increased preheating as well as duplicate IRSL readout at elevated temperatures as compared to classical IRSL measurements, sometimes including even additional hotbleaches), even if this is not necessary for the determination of the classical IRSL $D_e$ values or IRSL $g$-values. This extra heat input will make it even more difficult to separate "true" anomalous
fading from the decrease in signal intensity caused by dropping IRSL readout temperatures.

The magnitude of the unwanted effects depends on the particular measurement setup and protocol (e.g., type of luminescence reader, IRSL-readout temperature, preheat procedures, number of aliquots measured). Therefore, $g$-values determined with the SAR protocol on different luminescence readers in different laboratories with varying measurement protocols are *per se* not comparable among each other. This finding compares well with an interlaboratory trial on the kinetic parameters related to the
655 110 °C TL peak of quartz, where accurate temperature control was also found to be a major problem in standard luminescence readers (Schmidt et al., 2018). We conclude that the IRSL readout temperature after delay times of varying lengths after laboratory irradiation is most uniform if preheating is performed after the pause, i.e., immediately prior to IRSL-stimulation. Therefore, for SAR-based fading tests we recommend to follow the procedure of Rhodius et al. (2015), who deliberately placed the pause in between the laboratory irradiation and the preheating as not to produce erroneously high $g$-values. However, this
proposal may conflict with the concerns of Auclair et al. (2003) who assumed electron redistribution to occurr after preheating and disturb the fading measurements. Further, we recommend lowering the liftup temperature down to a level which considers the intended IRSL readout temperature appropriately, without elongating the total measurement time unduly by too long cooling-down times. We also recommend to repeat measurements with zero delay several times, both at the beginning and at the end of an entire fading measurement, in order to monitor the potential range of the scatter of the SAR values with zero
delay as well as potential overall trends or fluctuations during a complete fading measurement. If thresholds of recycling ratios of ± 10 % are accepted for the SAR $D_e$ measurements, which is usual practice in luminescence dating, it seems reasonable to expect similar scatter ratios in the frame of corresponding fading tests. Against such a background, the evaluation of potential data decline, due to fading of only few percent is a considerable, error prone, challenge in luminescence dating studies.



We consider it further necessary to include in a SAR-based fading test a series of very short (seconds to few minutes) and short pauses (minutes to few hours) to investigate whether malign effects of heat accumulation, as indicated by an initial plateau of the SAR values, are present or not. If such a plateau may be observed, the *g*-values will be likely overestimated and thus will be any fading-corrected IRSL ages based on these *g*-values. In that case the true age of a sample will range somewhere between the age based on the $D_e$ of a sample (minimum age) and the fading corrected age (maximum age). This interpretation may explain why fading corrected data (following, e.g., the procedure of Huntley & Lamothe 2001) may overestimate the true ages (e.g., Trauerstein et al. 2012).

If we finally assume that the linear part of the fading curve as displayed on a logarithmic time scale reflects the fading component, which is responsible for long-term signal decay leading to age underestimation of archaeological or geologic samples, then this linear part needs to be well identified and extracted for *g*-value determination from the overall fading curve. However, it is only by narrow spacing of pauses that a truly linear part of the fading curve may be determined. In the case of the present study, for the longest pauses of a few hours most measurements showed a declining slope, if not a bottoming out part, following the steep linear decline, next to an initial (semi-)plateau preceeding the steep (decreasing) linear part. Therefore, only a limited part of the fading curve appears to be appropriate as a base of *g*-value determination. However, in order to determine this part of the fading curve, many other fading dose points need to be measured in a time-consuming process which will be eventually discarded for the *g*-value determination. As an alternative, and for everyday dating applications likely a more practicable solution, few fading dose points may be measured. Yet, in agreement with the findings of Visocekas (1985) the shortest pause needs to last at least one magnitude of order longer than the irradition time. As, however, one cannot be sure whether or not the few measured dose points represent a linear function the ages calculated with the determined *g*-value should be regarded only as likely maximum ages, whereas the uncorrected ages could be regarded as likely minimum ages. Presently, with so little knowledge of the true nature of anomalous fading plus the interfering technical parameters during a SAR measurement this "soft" or "qualitative" approach seems to be an accepptible interim solution until the findings of the present study are better understood.

**Data Availability**

The data may be obtained by contacting the first author.

**Author Contributions**

AK performed the study and prepared the manuscript with input from all co-authors.

**Competing interests**

The authors declare that they have no conflict of interests.

**Acknowledgements**

Jutta Asmuth helped with preparing the aliquots for measurement. SK received funding by the LaScArBx LabEx, a programme supported by the ANR – No. ANR-10-LABX-52 and from the European Union's Horizon 2020 research and innovation programme under the Marie Skłodowska-Curie grant agreement No. 844457.

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

**Figures**



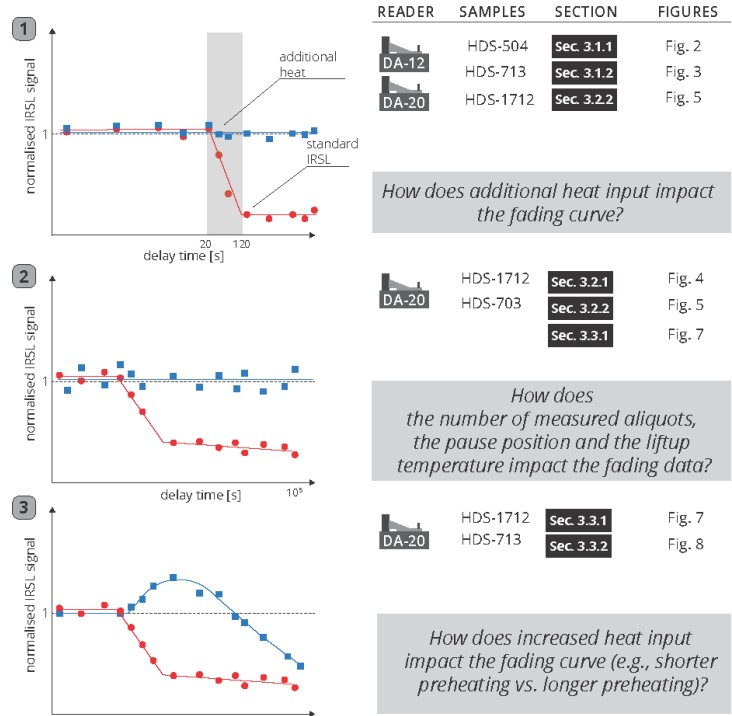

**Figure 1:** Overview sketch: Impact of SAR measurement parameters on the course of the fading data ('fading curve'). Few further aspects, such as, e.g., the number of measured dose points, different modes of nitrogen use and the importance of repeatedly measured zero-dose points both at the beginning and at the end of a measurent sequence, which are also addressed in sections 3 and 4, are not considered in this sketch.





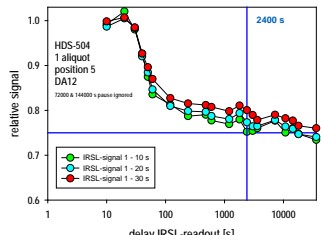

**(a)**

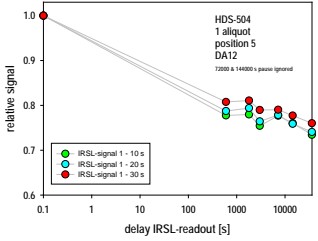

**(c)**

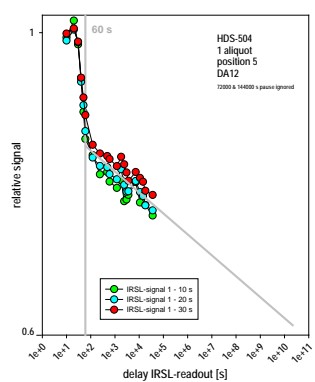

**(b)**

**Figure 2:** T*fad*-1, HDS-504 on DA12. **(a)** Results and **(b, c)** erroneous interpretations. Laboratory dose 5.2 Gy (180 s beta irradiation time), normalisation dose 2.6 Gy (90 s beta irradiation time). Preheat 120 s at 220 °C, IRSL at room-temperature, liftup-temperature 60 °C. **(a)** x-axis on a logarithmic scale. Blue lines indicating that the relative signal starts to bottom out after ca. 2400 s delay time. Initial zero-delay point not presentable on logarithmic scale. **(b)** x- and y-axis on a logarithmic scale. Grey lines indicating a breach in the slope after ca. 60 s delay time and a potentially (graphical extrapolation) considerable signal loss after a few $10^2$ ka. **(a, b)** Zero-delay point not viewable on a logarithmic scale. **(c)** Like (a), but thinned set of data points. Zero-delay point moved to 0.1 s delay time for reasons of presentability.




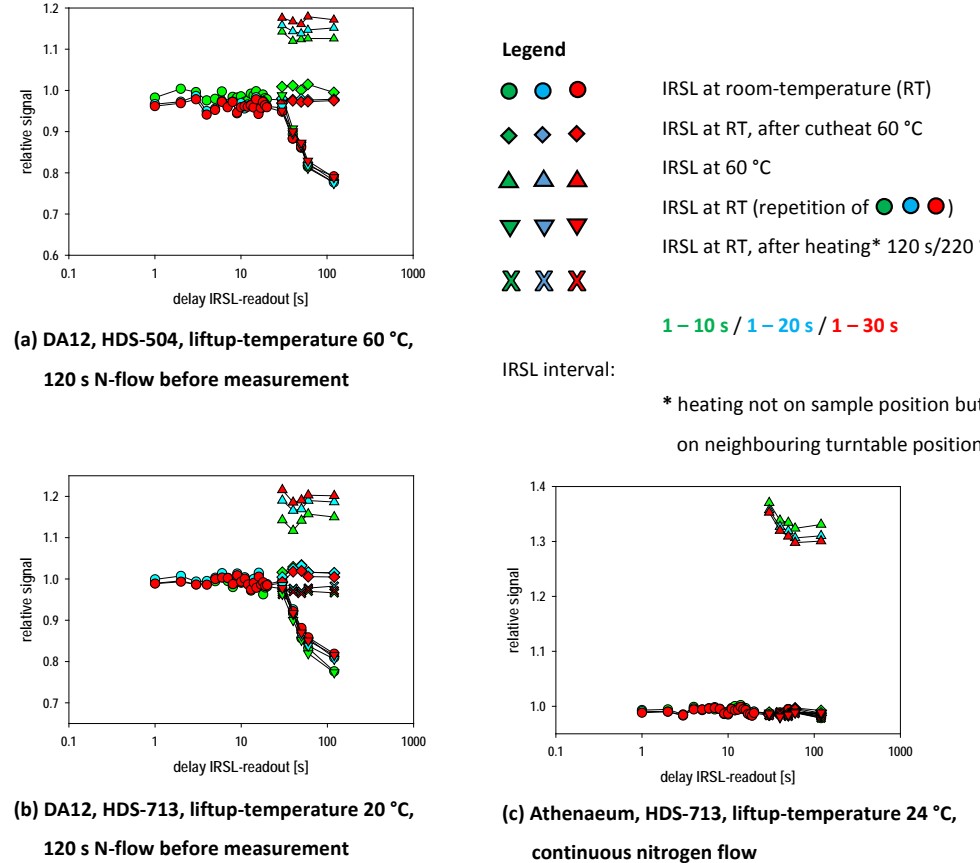

**Figure 3:** Measurements composed of **(a)** four or **(b, c)** five different subtests: IRSL readout at room-temperature (RT) without additional heat input (circles, down-pointing triangles) *versus* IRSL at room-temperature with additional heat input (diamonds, cross symbols) and IRSL at 60 °C (up-pointing triangles). **(a)** $T_{fad}$-2, HDS-504 on DA12. Subtest "IRSL at RT after heating 120 s at 220 °C" (cross symbols) not included in (a). **(b)** $T_{fad}$-3, HDS-713 on DA12 and **(c)** $T_{fad}$-4, HDS-713 on DA20 (Athenaeum). **(a)** Laboratory dose 5.2 Gy (180 s beta irradiation time), normalisation dose 2.6 Gy (90 s beta irradiation time). **(b)** Laboratory dose 4.6 Gy (180 s beta irradiation time), normalisation dose 2.3 Gy (90 s beta irradiation time). **(c)** Laboratory dose 17.3 Gy (180 s beta irradiation time), normalisation dose 8.6 Gy (90 s beta irradiation time). **(a – c)** Duration of beta irradiation 180 s. Preheat 120 s at 220 °C, IRSL at room-temperature. But: Different liftup temperatures. $T_{fad}$-3 was repeated with continuous nitrogen flow and $T_{fad}$-4 with only 120 s nitrogen flow at the beginning (results not shown here), with both tests giving same results as the here presented tests. **(a)** Circles (IRSL at room temperature) and down-pointing triangles (IRSL at room-temperature, repeat measurement) showing sudden and strong decline for delay times ≥ ca. 20 s. **(b)** Cross-symbols (IRSL at room-temperature after heating on a neighbouring turntable position) not showing the decline for delay times ≥ ca. 20 s. **(c)** Up-pointing triangles (IRSL at 60 °C) on reader DA20 showing downward trend not observed on reader DA12 (cf. this figure, a, b). Data of other subtests on Athenaeum scattering evenly around unity (individual subtests due to tight clustering of symbols almost not discriminable).





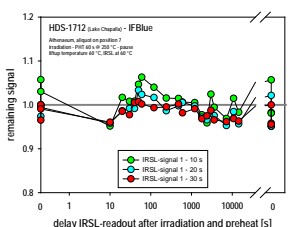

**(a)**

**Figure 4:** Results for HDS-1712 on DA20 (Athenaeum) with preheat 60 s at 250 °C, IRSL readout at 60 °C, liftup temperature 60 °C and irradiation, preheat, pause (Auclair et al. 2003). Laboratory dose and normalisation dose both 41 Gy (400 s beta irradiation time). **(a)** T$_{fad}$-5 (only one aliquot measured; max. delay 14400 s/4 h) *versus* **(b – d)** T$_{fad}$-6 (three aliquots measured in one sequence; max. delay 36000 s/10 h).

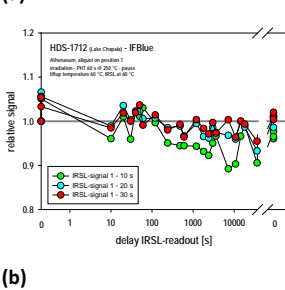

**(b)**

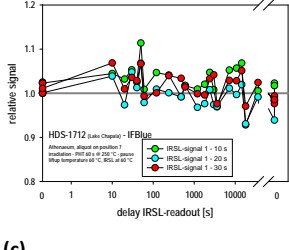

**(c)**

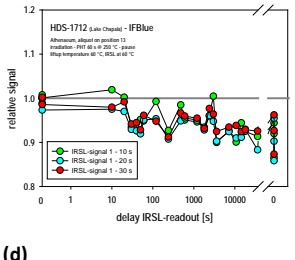

**(d)**

845





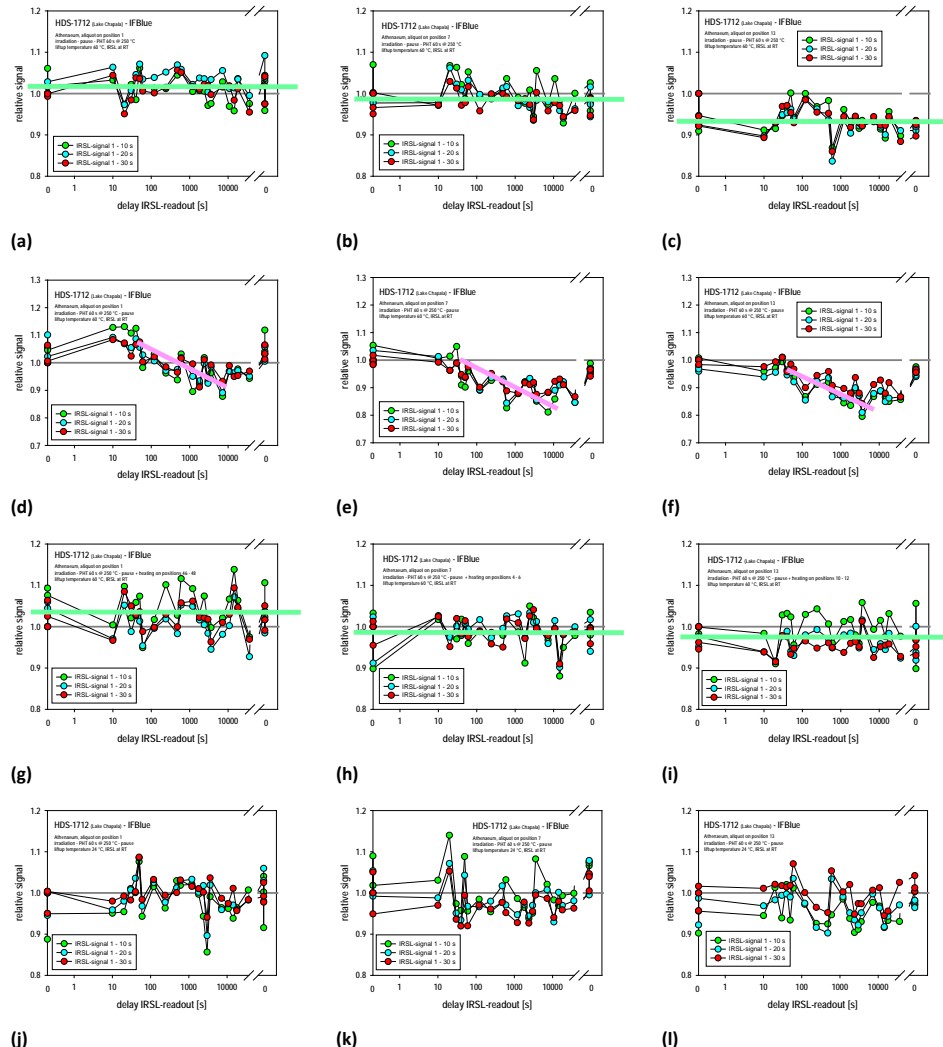

**Figure 5:** Results for HDS-1712 on DA20 (Athenaeum) with preheat 60 s at 250 °C and IRSL-readout at room temperature. Laboratory dose and normalisation dose both 41 Gy (400 s beta irradiation time).

- **(a-c)** $T_{fad}$-7, **(d-f)** $T_{fad}$-8, **(g-i)** $T_{fad}$-9, **(j-l)** $T_{fad}$-10.
- **(a – c)** Irradiation, pause, preheat (Rhodius e al. 2015) *versus* **(d – l)** Irradiation, preheat, pause (Auclair et al. 2003).
- **(a – i)** Liftup temperature 60 °C *versus* **(j – l)** liftup temperature 24 °C
- **(g – i)** Extra heating on neighbouring turntable position





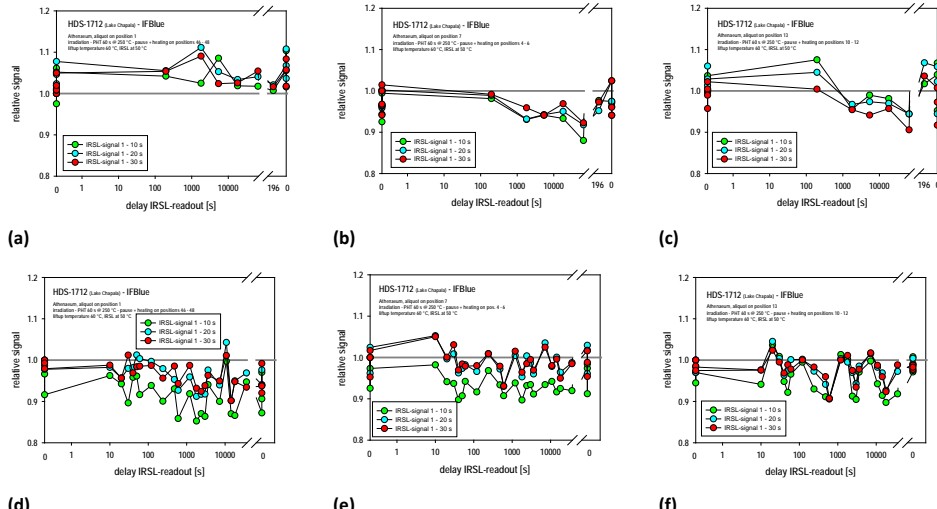

**Figure 6:** Results for HDS-1712 on DA20 (Athenaeum) with preheat 60 s at 250 °C, IRSL readout at 50 °C, liftup temperature 60 °C and extra heating on neighbouring turntable positions (3 x 60 s at 250 °C). Laboratory dose and normalisation dose both 41 Gy (400 s beta irradiation time). **(a – c)** Thinned out number of data points ($T_{fad}$-11) *versus* **(d – f)** increased number of data points ($T_{fad}$-12).

850





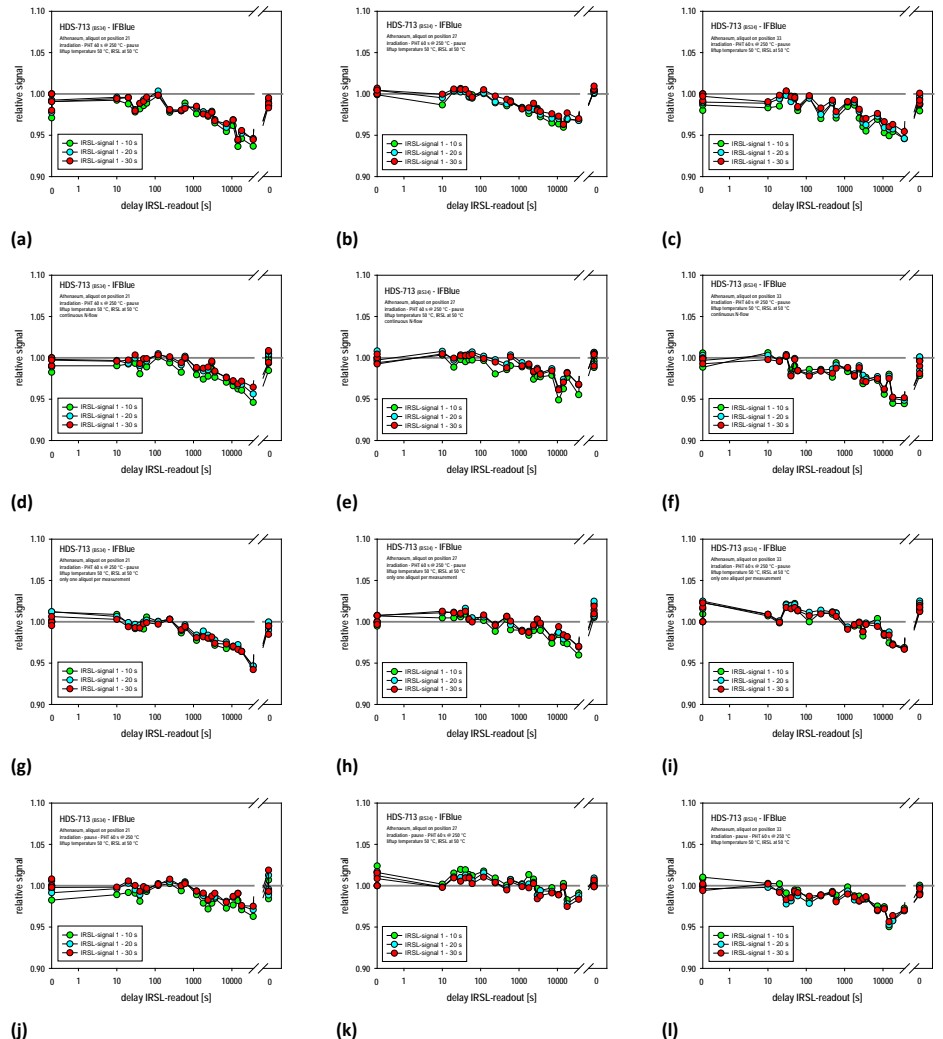

**Figure 7:** Results for HDS-713 on DA20 (Athenaeum) with preheat 60 s at 250 °C, IRSL readout at 50 °C, liftup temperature 50 °C. Laboratory dose 10.3 Gy (100 s beta irradiation time) and normalisation dose 5.2 Gy (50 s beta irradiation time).

- **(a – c)** only 120 s nitrogen flow at the beginning (T$_{fad}$-13)
- **(d – f)** continuous nitrogen flow (T$_{fad}$-14)
- **(g – i)** each aliquot measured separately in three individual sequences (T$_{fad}$-15)
- **(j – l)** pause before preheat (Rhodius et al. 2015) (T$_{fad}$-16)

For a comparison of the gross signal fading curve and the net signal fading curve cf. supplement 3, Fig. ii.



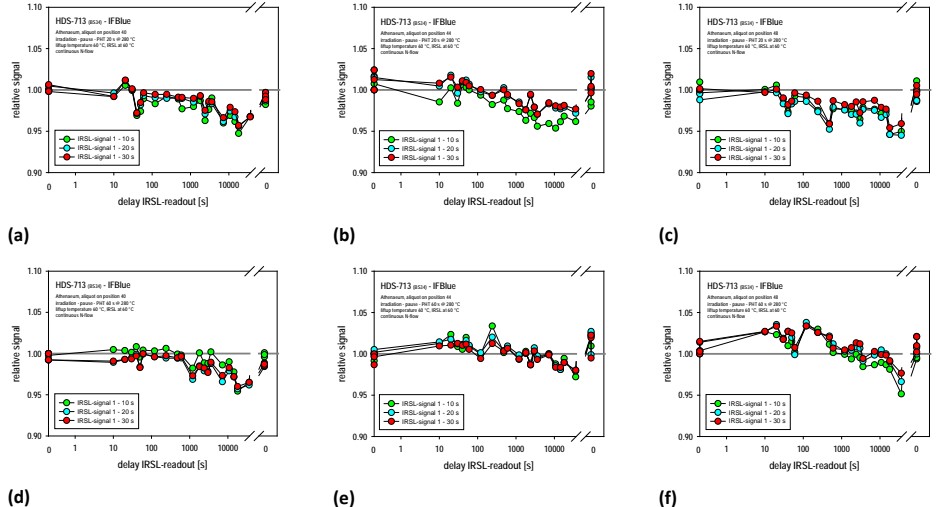

**Figure 8:** Results for HDS-713 on DA20 (Athenaeum) with IRSL readout at 60 °C and liftup temperature 60 °C. Laboratory dose 10.3 Gy (100 s beta irradiation time) and normalisation dose 5.2 Gy (50 s beta irradiation time).

- **(a – c)** Preheat 20 s at 280 °C (T$_{fad}$-17) *versus* **(d – f)** preheat 60 s at 280 °C (T$_{fad}$-18).

For a comparison of the gross signal fading curve and the net signal fading curve cf. supplement 3, Fig. iii.