# Peer review of "A closer look at IRSL SAR fading data and their implication for luminescence dating"

_Geochronology, 2020_

## Referee Comment (RC1) · Anonymous Referee #1 · 20 Mar 2020

The manuscript :

The manuscript under review presents a series of laboratory measurements intended to test the dependence of anomalous fading rate on the thermal environments of the IRSL measurements. Some of the conclusions deduced by the authors are shown graphically in the abstract; over short-term delay times they observe an increase of IRSL due to a delay in the cooling of the sample holder to reach the read-out T requested in the sequence (my understanding...), followed by a sharp decline in IRSL that is apparently tempering out at longer time delays. The authors thus propose that strict thermal conditions are required in AF studies and among other arguments would also suggest preheating irradiated aliquots after the pause in time necessary for assessing fading, which sits against the original protocol set up by Auclair et al.

[Figure]

Introduction to the review:

Before giving my appreciation of this manuscript, I would claim that I am a specialist in anomalous fading of feldspar luminescence. My group has carried out thousands of AF tests and we have been challenged quite a few times with scatter, lack of reproducibility, apparent trends between aliquots etc. Overall, we have at times observed plateaus or slight increases of IRSL during the early part of the AF experiments which we attributed to recuperation. Is this due to the thermal effects described here? Maybe but I do not think so (see comment below). However, this short-term malign behaviour seems to disappear for extended time delays and the rule of thumb is to try to stay away from very short delay measurements. This being said, let me remind every laboratory practitioner involved in feldspar luminescence that the assessment of g values involves measurement of minute luminescence intensity differences between aliquots and over an extended period of time. This is a very difficult task indeed.

The actual review

I must first admit that even though I know a lot about AF, I had to read the paper over and over again and I am not sure yet that I do understand exactly what the authors have been doing and why. In terms of format, the manuscript requires extensive rewriting and graphs revision before resubmission. The authors may wish to revise the document following the propositions below or at least provide arguments to support the claims carry by the paper. At this stage, I consider the manuscript should not be accepted for publication.

First of all, I have serious concerns about the scientific arguments of the manuscript.

1. When engaging in a scientific endeavour, one needs to have strong evidence that there is a large issue on hand that has been overlooked by the community. Here, there is a pervasive reference to a paper by Rhodius et al that would have shown unequivocally fundamental failures about the protocol of Auclair et al for measuring AF. Just for a quick reminder, the latter is one of the most referred papers in AF since its publication.

There are probably hundreds of publications in which this protocol has been used and found reliable. The rare papers that do not consider the AF g values measurements as accurate using Auclair et al are those dealing with high-temperature pIRIR (in which are small ca 1% g values reported. . .are they real?) and those for which applications of g values to the uncorrected ages results in age overestimations. It is not this reviewer job to judge these contributions but I know by experience that this is observed mostly in the context of possible partial bleaching of feldspar IRSL such as for fluvial and al-luvial sediments. In the case of the Rhodius paper, the measurements are carried out on rock slices, a difficult and relatively new application for which some aspects of age determinations could be questioned. 2. The proposal of Auclair et al to preheat just after the irradiation is to try to get the charge distribution as close to that of the natural as possible during storage. To preheat after storage thermally transfer electrons back in the dated trap. This is easily observed. Also, you may want to measure AF using short-shines so you need to preheat first as well (eg Huntley and Lamothe, 2001). 3. In science, if one wishes to test a protocol, the way to do so is to change only one variable at the time and compare results before and after a qualitative or quantitative change. We are faced here with an arsenal of methodological changes that do not allow the reader to make his own idea about the results of anyone experiment. In that sense, the methodology is not designed to allow drawing unequivocal conclusions. Therefore, please simplify the experimental tests and organize them so the argument of the ex-periment is structured. I understand that Figure 1 is intended to explain the readers just that but unfortunately, this figure does not seem to do the job. 4. If you wish to test new methodologies for assessing AF, you need 1) a monomineralogical sample, coarse grains K-feldspars in the best case; 2) a feldspar showing unequivocal fading, and 3) a bright thus highly dose sensitive feldspar. The samples used in this study are just not appropriate. IRSL emission from polyminerals may be from K-feldspar but also from albite or from another low-K feldspar even maybe from clay minerals. Sam-ples are dim except for sample 713, a coarse grains K-fds extract shown on figures 7 and 8. The log decay is very clear therein and there is no short-term increase. . ..

5. For the three stages of luminescence decays: plateau or increase, then decrease then plateau again...there are several examples of anomalous fading decay curves you could find in the literature (Huntley and Lamothe, 2001; Huntley and Lian 2006 and several others...) for which there is absolutely no evidence for cessation of fading. From the early work of Wintle and Visocekas to the more recent contributions of Huntley (see his paper in 2006 about the t-1 law), the dependence of the decay of luminescence intensity over the log of time has been clearly demonstrated. Therefore these three stages if observed are due to hardware-based temperature variability for the first part and absence of long storage times for the fading "flattening"...or the decline could be an artifact for a non-fading feldspar...whatever the cause, this fading structure has nothing universal. There could be contexts in which one may have issues in detection of AF decay as the non-fading component becomes dominant but fading continues... There is no such thing as the nonsensical expression "expiring of fading" as written on line 450; quantum mechanical tunnelling does not take breaks... Among technical changes that are requested: 1. Some sentences are difficult to understand due to poor English grammar (eg lines 350 to 355 as an obvious example). 2. The experiments are difficult to follow because the authors have decided to use acronyms of their own for describing their protocol. The use of terms such as LAB or NRM for normal Lx and Tx is very confusing. I would also require that the authors use Lx/Tx as the y axis instead of relative intensity (it is a sensitivity-corrected signal) for most of their figures. Use delay instead of pause (or storage). 3. Every figure should show only one stimulation time range (0-10 sec is fine), the use of three time ranges crams the graphics. 4. Each value needs to be properly calculated by subtracting the early part of the shine down by the late light, as done everywhere else. I do not get here the argument of why one decides to change a universal measurement protocol. 5. For the x axis, you are required to use a log of (time/tc) in which tc is approximately the half irradiation time plus the time between irradiation and measurement. You then get the zero point right and should be able to properly test decay log-linearity. 6. The extra-heating on another position: why? To get the heating plate always a bit hotter for a delay as it would be for a prompt

measurement? Then is this not a problem of hardware as the thermocouple is not doing its job? 7. The experiment of having a read-out temperature lower than the lift-up T may be interesting but this is not a routine way to measure luminescence, contrary to the impression we get from reading this part of the manuscript. 8. It is not a good idea to make any luminescence measurements at room temperature as this T may be different from day to day, hours to hours, minutes to minutes. . .always use some higher than RT temperature to measure IRSL. We use 50C, some have used 32C in my lab for a while, the idea is to control the temperature. . . 9. The conclusion for figure 1b is that extrapolation to extended times results in a large reduction of the signal. . . is this a problem? This has been known for decades (see Visocekas for extrapolation to the age of the Earth). . .the argument normally is that after some time the decay from the lab dose is the same as that in the field. . .reaching thus a state of quasi-equilibrium (see Lamothe et al 2003). 10. You need to properly refer to those early workers who have observed the relation between temperature and IRSL emission in feldspar, Bailiff and Poolton, Duller . . .this property has been known for some times. Along the same lines, it is a common practice in the Auclair et al protocol to run several prompt measurements to fix clearly the zero point on the (log of time) x axis. Our lab and others in the world have been doing this in routine for years. I would ask the authors to remove their strange claim that they have discovered this procedure. 11. I should point out that if you measure fading on a set of MAA, you need to subtract first the natural signal. . .I cannot see if this was done in the papers for which there was some "problems" (Lang, Rhodius, Kadereit. . .).
* * *

---

## Short Comment (SC1) · 25 Mar 2020

Sebastian Kreutzer

sebastian.kreutzer@aber.ac.uk

Dear Anonymous Referee,

I appreciate the thorough and critical comments on our manuscript, to which I may respond as part of the open discussion encouraged by Geochronology.

When Annette approached me with early results in 2017 that would later be part of this manuscript, it became, very soon, clear to me that a publication of these results will become tricky. But still, I am happy to see these challenging comments, because what we need is a proper discussion and an exchange of arguments.

You open the comments with a request for an "extensive rewriting and revision of graphs". This is a bold statement, and it indicates that we somehow failed to lay out

reasoning and results convincingly in the reviewer's eyes, and we will certainly try to do better. Putting this aside, I have to admit that I sometimes struggled with these rather general comments. They read clear and convincing in the first place, but sometimes they lack the link to our manuscript and our results. Nevertheless, I tried to address your concerns with this quick response so that we may clarify some points before we prepare an improved version of the manuscript.

**1 Response to (1)**

I found the comments from that point very difficult to address because they do not refer to the results presented in our manuscript. The bottom line seems to be that you are unhappy with the repeated reference to Rhodius et al. (2015) while the paper by Auclair et al. (2003) is *"Just for a quick reminder, the latter is one of the most referred papers in AF since its publication."* I do not intend to play down the impact of the work by Auclair et al. (2003). It was, and is, a solid piece of work, a fact which is not questioned by our work. Do you expect us to add something in particular to the text?

**2 Response to (2)**

I am not sure whether I got the point. Do you want us to discuss the possibility of thermal transfer? You refer to Auclair et al. (2003) who wrote that *"g-values that are underestimated, probably as the result of thermal transfer."* This is something we mentioned very early in the manuscript in Sec. 1.4 and, of course, later before Sec. 4.2.

**3 Response to (3)**

Here you wrote that *"In that sense, the methodology is not designed to allow drawing unequivocal conclusions."* I admit that I can understand that the presented experiments appear rather dense, and perhaps we should try to simplify the writing further. But the here made general statement does not provide much guidance.

For instance. In Sec. 3.1 the header states that we compare "IRSL readout without vs. IRSL readout with thermal assistance". We then present results from experiments for a DA12 and a DA20 luminescence reader (Sec. 3.1.1 and Sec. 3.1.2). Sec. 3.1.1 is structured as follows:

1. We provide a context for the loess sample used in our experiment

2. We point out that, with the same sample MAA and SAR shows different behavior in terms of measured fading

3. We illustrate the effect of different pause (delay) times

4. The we apply IRSL at room temperature and with thermal assistance (heating)

5. We present the results

Again, I admit that it takes time and needs a careful reading of the text to follow the experimental design. So, of course, we are happy for any hint to improve the readability without neglecting essential technical details.

**4  Response to (4)**

Your comment claims that our samples are just not appropriate for our experiments. This argument, however, twists the aim of our manuscript. What we did was to test and question the way fading is commonly measured. We questioned whether the SAR approach is the most appropriate way to do so, given our results, and measured with the samples at hand. We did not examine fading and we did not claim that our results change underlying physical assumptions.

One could say that we used the wrong samples, and because of this, we measured what we measured. But what does it imply for all the fading measurements out in the wild using 'non-perfect' samples? What we found, we found highly reproducible, and you are right, maybe this is all just related to our samples, and our measurement conditions caused this effect. But still, the impact is real and would bias fading measurements in the way they are currently done.

> *"Samples are dim except for sample 713, a coarse grains K-fds extract shown on figures 7 and 8. The log decay is very clear therein and there is no short-term increase"*

Dim does not mean unsuitable and should not be understood that way. More important, only the Mexican lake sample (HDS-1712) was a rather dim (Figs. 5 6), and all samples are fine-grain samples.

A plateau would be best visible with more data points at the beginning, which was not the case for this experiment. However, you can still see a plateau and an overshooting (e.g., Fig. 8f). However, we will try to make this clear in the figure caption to avoid this misunderstanding.

**5   Response to (5)**

*For the three stages of luminescence decays: plateau or increase, then decrease then plateau again...there are several examples of anomalous fading decay curves you could find in the literature (Huntley and Lamothe, 2001; Huntley and Lian 2006 and several others. . .) for which there is absolutely no evidence for cessation of fading.*

We did not question this in our manuscript.

*From the early work of Wintle and Visocekas to the more recent contributions of Huntley (see his paper in 2006 about the t-1 law), the dependence of the decay of luminescence intensity over the log of time has been clearly demonstrated. Therefore these three stages if observed are due to hardware-based temperature variability for the first part and absence of long storage times for the fading "flattening". . .or the decline could be an artifact for a non-fading feldspar. . .whatever the cause, this fading structure has nothing universal.*

This is somehow similar to my response above and fully in line with our argumentation: Whatever it is, it has nothing to do with 'true' fading.

*There could be contexts in which one may have issues in detection of AF decay as the non-fading component becomes dominant but fading continues. . . There is no such thing as the nonsensical expression "expiring of fading" as written on line 450; quantum mechanical tunneling does not take breaks.*

I do agree and we did not claim this, we just repeated observations and this has something to do with the way it gets measured and is not related to 'true' fading.

**6   Technical issues**

These are valuable comments and we will address them separately in more detail. We understand that the experiments are difficult to follow, they are indeed complex and we will try to further simplify the text.

A few preliminary questions/comments here:

*I would also require that the authors use Lx/Tx as the y axis instead of relative intensity (it is a sensitivity-corrected signal) for most of their figures. Use delay instead of pause (or storage).*

Normalizing signals is useful to compare results with different $L_x/T_x$ values and nothing special. For the moment, I cannot see what we would gain from plotting $L_x/T_x$ values, except that it would be harder to compare the curves. Please correct me if I fundamentally overlook an aspect that needs to be pointed out by plotting $L_x/T_x$ values. The term 'pause' was chosen because it is a 'pause' that is requested in the sequence editor and not a 'delay' (which does not mean that it cannot be changed).

*Every figure should show only one stimulation time range (0-10 sec is fine), the use of three time ranges crams the graphics.*

This would be only meaningful for curves where the 1-10 s range matters while the rest does not.

*Each value needs to be properly calculated by subtracting the early part of the shine down by the late light, as done everywhere else. I do not get here the argument of why one decides to change a universal measurement protocol.*

What would be *'everywhere else'* and *'universal measurement protocol'*. To avoid mis-understanding, I am sincerely not sure what is meant here. Do you want to imply that there is something wrong in particular with our results? What would (probably) change?

> *For the x axis, you are required to use a log of (time/tc) in which tc is approximately the half irradiation time plus the time between irradiation and measurement. You then get the zero point right and should be able to properly test decay log-linearity.*

I do remember that we discussed this, and I even had in mind that we double-checked this, but it does not change the results. Anyway, of course, I will recheck it with the other authors.

> *The extra-heating on another position: why? To get the heating plate always a bit hotter for a delay as it would be for a prompt*

No, as written in the text, to have thermal assistance due to, if you want to say so, 'cross-talk' of the heating. It is a rather strange experiment, I do agree, but it does not make a claim on the thermocouple.

> *The experiment of having a read-out temperature lower than the lift-up T may be interesting but this is not a routine way to measure luminescence, contrary to the impression we get from reading this part of the manuscript.*

This experiment was performed to show the influence of possible heat assistance, varying/decaying with delay time. We will clarify this in the text.

*It is not a good idea to make any luminescence measurements at room temperature as this T may be different from day to day, hours to hours, minutes to minutes. . .always use some higher than RT temperature to measure IRSL. We use 50C, some have used 32C in my lab for a while, the idea is to control the temperature.*

The temperature in the room was controlled, given its location in the basement, and stable during the time of the experiments. We cannot exclude minor temperature differences though.

*9. The conclusion for figure 1b is that extrapolation to extended times results in a large reduction of the signal. . . is this a problem? This has been known for decades (see Visocekas for extrapolation to the age of the Earth). . .the argument normally is that after some time the decay from the lab dose is the same as that in the field. . .reaching thus a state of quasi-equilibrium (see Lamothe et al 2003).*

I don't see a problem, but we will discuss this. However, the result was striking, because Lang et al. (1996) could produce MIS 2 – MIS 5 IRSL-ages which were in correspondence with the $^{14}$C ages (of course where such a match was possible). If fading would be have been an issue such results would have been unlikely. Maybe something else went wrong, and they got the results in agreement by chance, but for obvious reasons, such an assumption would not be a sensible approach.

*10. You need to properly refer to those early workers who have observed the relation between temperature and IRSL emission in feldspar, Bailiff and Poolton, Duller . . .this property has been known for some times. Along the same lines, it is a common practice in the Auclair et al protocol to run*

*several prompt measurements to fix clearly the zero point on the (log of time) x axis. Our lab and others in the world have been doing this in routine for years. I would ask the authors to remove their strange claim that they have discovered this procedure.*

Of course, we should adequately cite previous work, and we will recheck this part. To my understanding, we did not claim that we have discovered this but pointed this out (again). The problem with these 'common' procedures that may have been used or not in different labs is all the same: I do agree, many have seen this problem before, indeed, this is nothing new, but we should discuss what does it mean for the current approach to measuring fading.

*I should point out that if you measure fading on a set of MAA, you need to subtract first the natural signal. . .I cannot see if this was done in the papers for which there was some "problems" (Lang, Rhodius, Kadereit. . .).*

Rhodius et al. (2015) did not apply MAA, but SAR. Further, they used the Auclair et al. (2003) protocol, but also a variant of it. Presented in their publication are both results. For the other articles, we refer to our supplement (supplement 1) were presented the MAA fading test procedure applied by Lang et al. (1996) who had published their papers in renowned peer-reviewed journals. For constructing the MAA dose-response curve you always use the laboratory dose on top of the natural dose. It does not seem mandatory to first subtract the natural signal. Although this might lead to possible stronger fading (theoretically), the subtraction would lead to larger errors following Gaussian error propagation, which in turn could camouflage true fading. There seem to be pros and cons to this issue, which, however, is not the focus of the present manuscript.

We are, I guess I can speak for all authors of this manuscript, looking forward to continuing this discussion.

Best regards,

Sebastian Kreutzer

**References**

Auclair, M., Lamothe, M., Huot, S. 2003. Measurement of anomalous fading for feldspar IRSL using SAR. Radiation Measurements, 37, 487–492, doi:10.1016/S1350-4487(03)00018-0, 2003.

Lang, A., 1994. Infra-red stimulated luminescence dating of Holocene reworked silty sediments 13, 525–528.

Rhodius, C., Kadereit, A., Siegel, U., Schmidt, K., Eichmann, R., Khalil, L.A., 2015. Constraining the time of construction of the irrigation system of Tell Hujayrat al-Ghuzlan near Aqaba, Jordan, using high-resolution optically stimulated luminescence (HR-OSL) dating. Archaeol Anthropol Sci 9, 345–370. doi:10.1007/s12520-015-0284-x

---

## Referee Comment (RC2) · Anonymous Referee #2 · 4 Apr 2020

This paper reported some experimental observations about anomalous fading using two polymineral samples. The authors claimed that they have found different observations from what the majority of other people did, and provided comprehensive interpretation on their observations, and suggested a different way of doing fading tests. Although I understand that the authors spent a large amount of efforts in presenting the data and providing interpretations, I feel frustrated to try to understand and follow what exactly the authors have done. The manuscript was written and organised in a complicated way, which prevents a good flow of the contents. If I understand the authors correctly, I am afraid of that their observations are built on a wrong way of data presentation, so most of their explanations and implications for dating are not supported.

**Major issues:**

1) One of the major issues is that the authors presented their fading data in a wrong way. They artificially set the delay time of the first measurement (prompt measurement) to zero. This will artificially change the shape of the fading curve (and fading rate). Let's take a simple example. Suppose that we have a sample with a fading rate of 3 %/decade, and we conducted a series of delayed SAR measurements with irradiation (400 s). The IR-readout or delay time after the mid-point of irradiation is 300, 310, 350, 400, 500, 600, 800, 1000, 2000, 4000, 10000, 50000, 100000 s, respectively, which corresponding to about maximum of 2.5 decades of time (similar to most of fading tests conducted in previous studies). Supposing that its fading follows a logarithmic decay, we can theoretically calculate their decay curve (see the blue dots in the following figure). However, if we artificially set the initial decay to 1 s (btw, I don't know how could the authors show the first data point in a log scale, if the first time is 0 s, do you just ignore the first data in the figures?), what happens is that the decay curve (orange triangles) is flatten out at short delay times (

2) If my above comment is built on misunderstanding of the authors way of presenting data, and supposing that the author's presentation is correct (e.g., if we can indeed measure the fading curve down to a few seconds after irradiation, e.g., assuming that we can apply a short pulse of large dose), then do we expect such a pattern for real sample? The answer is yes, and fading curve (signal loss) should not follow a logarithmic decay function. The theoretical decay of signal should actually follow a power-law as proposed by Huntley (2006) in their Figure 2 (see below). If we compare the theoretical curves with the patterns shown in this study, they are remarkably similar, with a flat decay followed by a sharp decay and then flat decay again. The reason why we always assume a logarithmic decay (time too short) and the end of decay (storage time too long). The middle part of the power-law decay is what we can practically observe, and it follows roughly logarithmic decay.

3) Should we apply preheat immediately after irradiation or after delay for fading test? My answer is "both are incorrect but a prompt preheat is definitely better". We already know that, fading rate is strongly dependent on the charge concentration and distribution in the lattice (e.g., Huntley 2006; Li and Li, 2008). A prompt preheat is the best way to mimic the electron-hole distribution for natural samples, whose thermally unstable (and athermally unstable) electron-hole traps remain empty. Of course, one would expect even larger fading in natural process, because the electrons can undergo multiple filling and escaping process (hence more fading), whereas they only have one-off filling and escaping in the lab fading test (because irradiation time is much shorter than storage time, and the signal loss can only be monitored after irradiation). So, the authors' claim of "preheating prior to IRSL-readout of the natural dose occurs after a long storage in nature" is wrong. Natural process is a long-term 'preheat' at low temperature. Kinetically, it has a similar effect of short-term preheat at

high temperature. That's why we need to apply preheat in the laboratory to mimic such process. Fading test, therefore, should not be conducted by keeping these unstable traps filled (e.g., using delayed preheat).

4) The authors discussed the fading test results without considering the fact that De measurements usually have the same parameters used for fading tests, such as stimulation temperature, and the magnitude of delay time between irradiation and IR measurements. So any effect of changing in measurement conditions for fading test, may also influence the De estimates. For example, let's take two scenarios. The first scenario is that both fading test and De were measured by using room temperature IR stimulation, and the 2nd scenario is that they were measured at 60 degree C. If one get a higher g-value when room-temperature IR readout is applied, then the corresponding De would probably lower too. That means, one can probably get the same fading-corrected results no matter what readout temperature they use and what fading rates they get. Of course, whether fading correction is reliable is another topic, as we need a model to describe the fading process, which is poorly known unfortunately.

This issue also applies to the other experiments described by the authors, e.g., heat input, pause position, liftup temperatures, etc. Ultimately, the authors should test whether these conditions also influence De results or not. If they do, will you get the same results after fading correction? If you do get the same results, then that suggest no matter what fading rates you get, they are reliable.

Minor issues:

- Section 1.6. The authors used some example of pIRIR results to support that fading measurement cannot provide insight to the stability of pIRIR. However, they ignored the fact that there are many other effects contributing to whether a pIRIR age is consistent with 'expected age' or not, such as residual correction (see the recent paper by Rui et al. QG, 2020), initial sensitivity changes (e.g., Qin et al. QG; Zhang et al., QG). They also ignore that not all the pIRIR results are reliable (see Li et al., 2014 Geochronometria for a review). As the other reviewer suggested, if one wants to study the effect of fading, they need to do a factorial experiment, i.e., ensure that only one parameter (fading) can affect the results. But I doubt this is possible at this stage, given the complex processes involved in IRSL dating.
- 2) Section 2.2. I am confused about why the authors did not use the option "run one aliquot at a time". My understanding is that, if they chose not to use this option, then the timing between irradiation and IR readout would vary from aliquot to aliquot (if they input the aliquot positions in the same sequence for each row). Then this would cause problem in SAR procedure, as different aliquots would have different delay time between the test dose irradiation and test dose signal measurements, which can result in problems with sensitivity corrections using test dose signals (as the test dose signal will fade differently for different aliquots).
- 3) The authors conducted many different version of fading tests with different parameters. I found it is extremely frustrating to try to understand what exactly they have done by reading the texts (section 3). I would suggest that the authors

provide a flow chart or sequence table (such as those commonly used to describe SAR procedures) for each of the fading tests, which will greatly aid the readers to understand details about their experimental procedures.

- 4) Is NRM the same as test dose? If yes, please use test dose, to keep consistent with the term used commonly by others.
- 5) The authors' results are based on polymineral, which contains all kinds of feldspars. What happens if Na-feldspar, Ca-feldspar and K-feldspar fade differently and respond to different stimulation conditions differently? For example, if some of them has an extremely high fading rate (say, ~20%, then the overall effect is that the fading decay will be flatten out in the later part of delay as the fast-fading component become substantially small than stable components.
- 6) Figure 1. I have no clue what the data points or model represents. What do different colours and symbols represent in 1, 2 and 3?

---

## Author Comment (AC1) · 7 Jul 2020

**A closer look at IRSL SAR fading data and their implication for luminescence dating – Final Response**

Annette Kadereit[1], Sebastian Kreutzer[2, 3], Christoph Schmidt[4, 5]

[1] Heidelberger Lumineszenzlabor, Geographisches Institut, Universität Heidelberg, Im Neuenheimer Feld 348, 69120 Heidelberg, Germany

[2] Geography & Earth Sciences, Aberystwyth University, Aberystwyth, SY23 3DB, United Kingdom

[3] IRAMAT-CRP2A, UMR 5060, CNRS-Université Bordeaux Montaigne, Pessac, France

[4] Lehrstuhl Geomorphologie, Universität Bayreuth, Universitätsstr. 30, 95447 Bayreuth, Germany

[5] University of Lausanne, Institute of Earth Surface Dynamics, Quartier UNIL-Mouline, Bâtiment Géopolis, CH-1015 Lausanne

We thank the anonymous reviewers for their thoughtful comments. Basically, the reviewers argue that our observations may be owed to (1) the x-axis scaling in the figures and (2) the use of polymineral fine grains instead of (K-)feldspar coarse grains. Therefore, we re-analyzed our data, created graphs as commonly used for g-value[1] estimations and accomplished our SAR[2] IRSL fading measurements by SAR post-IR$_{1st}$IR$_{2nd}$ fading measurements, both on polymineral fine-grains and feldspar coarse grains. In short, the phenomenon of an initial (semi-)plateau exists and applies also to pIR$_{1st}$IR$_{2nd}$ as well as feldspar coarse grains. As pointed out by the reviewers, g-values from pIR$_{1st}$IR$_{2nd}$ measurements above zero may be measurement artifacts (cf. also Thiel et al. 2011). Our pIR$_{1st}$IR$_{2nd}$ measurements give hints on how such artifacts could possibly be generated. However, during the data reanalysis we also learned that the steering software of the luminescence reader behaved in an unexpected way and that therefore we cannot stick to the original interpretation that the shape of the fading curve may be attributed to varying heat assistance. For this reason, we have to withdraw our manuscript. Nevertheless, we would like to share our experience, so that nobody else will repeat the same mistake (section 2). Further, we share some of our reanalyzed IRSL data (section 3) and newly acquired pIR$_{1st}$IR$_{2nd}$ data (section 4). We will start with a consideration on the graphical presentation (section 1).

**1 Plots and x-axis scaling**

As our observations had made us suspicious of the data curves we did not calculate any g-values from them in the manuscript, as this did not seem appropriate. We decided to leave the data as unprocessed as possible. This is why we used gross signals instead of net signals and merely ensured that both signals follow the same trend. From our perspective now, this procedure was crude but ok. But we also used a rough approach for the scaling of the x-axis. Instead of plotting decades $(\log_{10}[t/tc])$[3] we plotted the „pauses" as typed in the sequence editor on a logarithmic scale. This allowed a better visual inspection of the data points of the short pauses which are too cramped and undecipherable otherwise (**Fig. 1a** vs **1b**). In order to compare the initial prompt readouts with the final prompt readouts we plotted the first close to zero, as explained in the manuscript, and the latter after a breach in the x-axis. As we did not perform any calculations, modelling or other mathematically-based analyses on the data and as we did not
* * *
[1] denoting the percentage fading loss of a luminescence signal per decade (e.g., Aitken 1985)
[2] Single aliquot regenerative (Murray & Wintle 2000)
[3] tc denoting the prompt readout and t denoting any delayed readout

want to overload the manuscript with more graphs (zooms, insets) we had decided for this „shortcut".

Our measurements were originally carried out with an old Risø TL/OSL DA12 system with a software emulator. In this system, the BIN-file does not provide information on the „time since irradiation". Thus we used the pause-times from the sequence, later also for data obtained with the DA20 reader. It is obvious, that this crude approach neglects the offset of the „prompt" delay time, which includes half the irradiation time, time for cooling, moving of the turntable, heating, liftup and so forth. In the case of the SAR IRSL measurements of our study this delay time adds up to ca. 280 s. But as shown in **Fig. 1a**, which considers this offset (and the normalisation to the prompt readout (tc)), this does not significantly transform the shape of the fading curve. Please note that the graphs with a logarithmic x-axis in the final response do not show the zero-values, which is a clear disadvantage in view of a desirable quick optical inspection of the complete SAR measurement from the initial to the final prompt readouts. Nevertheless, the logarithmic scale allows for better optical inspection (e.g., identification of inflexion points) of the early part of the fading curve than a linear scale. The thin grey line connecting the data points serves as a guide for the eye.

[Figure]

**(a)**                  **(b)**

**Fig. 1** Results for sample HDS-713 on reader DA20 (Athenaeum) with preheat 60 s at 250 °C, IRSL readout at 50 °C, liftup temperature 50 °C. Laboratory dose 10.3 Gy (100 s beta irradiation time) and normalisation dose 5.2 Gy (50 s beta irradiation time). $T_{fad}$-15 (each aliquot measured separately in three individual sequences), aliquot 3. Time since irradiation ($\log_{10}$ [t/tc]) plotted on **(a)** a logarithmic scale and **(b)** a linear scale, as conventionally used for g-value estimation.

**2      An unexpected finding - pauses are not processed as expected**

Our original idea was to present data and a first visual interpretation, but not to calculate any g-values from them. However, after having received the reviews from GChron we re-analysed the data we had compiled in the manuscript with the function „analyse_FadingMeasurement()" of the R package „Luminescence" (developer version 0.9.8.9000-17, Kreutzer & Burow 2020). For illustrative purposes the data from R were further processed with SigmaPlot (v11.0). This time, however, we worked off the measurements in a reversed order, starting with those on reader DA20 which did provide the „times since irradiation". Surprisingly, the results showed in the beginning of each measurement a larger number of (up to ca. 7) data points for $\log_{10}$(t/tc) around 0 (indicating prompt readout) than would correspond to the three initial dose points associated with a „pause" (sequence editor) of 0 s. This means that data points which in the

original manuscript version had been plotted „prompt, prompt, prompt, 10 s, 20 s (and perhaps 30 s, 40 s) all represent „prompt" readouts. What could be the reason?

It appears that this was due to an unexpected behaviour of the reader software. We assumed that a "pause of x s" in a sequence essentially adds a "pause of time x s" to the measurement. This, however, was not the case as shown by the two following screenshots (**Fig. 2**). **Fig. 2a** shows the sequence screenshot for which all run cells are identical, except for row three where the pause increases from 0 s to 60 s. However, as **Fig. 2b** shows, the corresponding time since irradiation (right column) was always around 280 s and increased only with run 10, by 8 s, while the actually requested pause in run 10 was 50 s. Why is this so?

Several reasons are possible including that the data in the column "time since irradiation" (**Fig. 2b**) is not correct. Although this might be possible we do not regard this as likely.

| Run 4 | Run 5 | Run 6 | Run 7 | Run 8 | Run 9 | Run 10 | Run 11 |
|---|---|---|---|---|---|---|---|
| Beta 100s | Beta 100s | Beta 100s | Beta 100s | Beta 100s | Beta 100s | Beta 100s | Beta 100s |
| Pre Heat 250°C;10°C/s;60s | Pre Heat 250°C;10°C/s;60s | Pre Heat 250°C;10°C/s;60s | Pre Heat 250°C;10°C/s;60s | Pre Heat 250°C;10°C/s;60s | Pre Heat 250°C;10°C/s;60s | Pre Heat 250°C;10°C/s;60s | Pre Heat 250°C;10°C |
| Pause 0s | Pause 0s | Pause 10s | Pause 20s | Pause 30s | Pause 40s | Pause 50s | Pause 60s |
| | | | | | | | |
| OSL 50°C IR LEDs;240.00s;5°( | OSL 50°C IR LEDs;240.00s;5°( | OSL 50°C IR LEDs;240.00s;5°( | OSL 50°C IR LEDs;240.00s;5°( | OSL 50°C IR LEDs;240.00s;5°( | OSL 50°C IR LEDs;240.00s;5°( | OSL 50°C IR LEDs;240.00s;5°( | OSL 50°C IR LEDs;2 |

**(a)**

| POSITION | LTYPE | RUN | SET | TIMESINCEIRR |
|---|---|---|---|---|
| All | All | All | All | All |
| 21 | IRSL | 1 | 5 | 278 |
| 21 | IRSL | 2 | 5 | 279 |
| 21 | IRSL | 3 | 5 | 279 |
| 21 | IRSL | 4 | 5 | 280 |
| 21 | IRSL | 5 | 5 | 279 |
| 21 | IRSL | 6 | 5 | 280 |
| 21 | IRSL | 7 | 5 | 280 |
| 21 | IRSL | 8 | 5 | 280 |
| 21 | IRSL | 9 | 5 | 279 |
| 21 | IRSL | 10 | 5 | 288 |
| 21 | IRSL | 11 | 5 | 298 |
| 21 | IRSL | 12 | 5 | 395 |

**Fig. 2** Screenshots for comparing (**a**) the input in the sequence editor with (**b**) the processed measurement sequence. Here cut-out Run 4 to Run 11, Set 1 to Set 3.

**(b)**

Assuming that the data in the BIN-file are correct (**Fig. 2b**) we consider the following assumption as most likely: After preheating (or any other measurement step involving increased temperature) the reader needs some time to cool down to the set threshold of the liftup temperature. As however „pauses" (and here we can only speculate, as we do not know the source code) are likely not a step for which it is checked whether or not the liftup temperature has already been reached before a pause starts, pauses may become part of the idle time of the cool-down process. This way short pauses may become completely „used up" by the cool-down process (cool-down time > pause), and only longer pauses (pause > cool-down time) will effectively elongate the delay time. If this consideration is correct, the very early plateaus as presented in the manuscript are merely artefacts. These very early plateaus do not exist (hereafter „fake plateaus"). On reader DA20 the first (up to ca. 7) data points need to be transferred to 0 ($\log_{10}$ [t/tc]) on the x-axis (or 0 s „delay IRSL-readout" in the manuscript version). On reader DA12 up to ca. 20 data points are affected representing the very short pauses, which were in steps of 1 s.

Unfortunately, we were not aware of this unexpected software design, which made us investigate shorter and even shorter delay times, down to 1 s on reader DA12, and assume that thermal assistance of the IRSL readout after very short and short delay times causes the

emergence of an initial plateau in the fading curves. As this interpretation can not be kept up we can not proceed with a publication of our data in GChron but have to withdraw the manuscript.

Whatever the reasons for the unexpected processing of the input data in the sequence editor are and regardless of the fact that aliquots are not "lifted up" on the heating plate during "pauses", we would appreciate a software which treats all pauses in the same way.

Despite this drawback, our experiments have produced valuable data. Although the very tips of the initial plateaus, the "fake plateaus", disappear to condense into an unexpected large number of initial prompt readouts, the fading curves still exhibit an initial plateau (**Fig. 3f**).

**3       Calculating g-values from our SAR IRSL measurements**

For the g-value calculations with the R package „Luminescence" we used an early integral of 1–20 s and a late-light subtraction of 201–240 s. The graphs show colour-coded lines to indicate selected sections of the curves and the resulting g-values if those sections are used for g-value determination. Stars serve the same purpose. In those cases, we did not use entire sections of the curves, but only the points highlighted by means of the star symbols.

Our maximum delay times are too short for reliable g-value calculation. The numeric values, however, support the optical inspection of the shape of the fading curves. This way, the g-values serve as a proxy, similar to the numerical expression indicating the slope of a regression line.

The results are summarised as follows:

− Including three final prompt readouts serves to monitor the overal stability of the SAR measurement. In our case, including or excluding the final three prompt readouts for g-value estimation does not change the numerical results significantly. The g-value calculations appear quite robust in this respect.
− Most measurements seem to show an initial plateau, a less steeper gradient or a kind of flat-step stair („semi-plateau") up to ca. 0.5 decades, which is sometimes shorter and sometimes longer. In this part of the fading curve g-values may be smaller than for the subsequent part and/or for the complete data set, as indicated by few arbitrarily given examples in **Fig. 3** and **Fig. 4**.
− Generally, the numeric data (g-values) support the visual impression of an initial (semi-) plateau. This finding conforms to Visocekas (1985, 1993) and Huntley & Lamothe (2001) who exclude fading shortly after irradiation and to Auclair et al. (2003) who showed that effects of thermal electron transfer may overprint anomalous fading if preheating is performed immediately before the IRSL readout.
− As indicated by the examples supplemented by red star signatures (three initial prompt readouts, one data point towards the end of the initial plateau, two data points representing the two longest delay times): If only very few data points are used and one of these sits near the end of the initial plateau this may slightly increase the g-value as compared to the complete data set (grey and light blue lines and numbers). This corroborates to Huntley & Lamothe (2001) who argue that the log-time equation does not apply to very short times.

- This finding also confirms our concern – expressed in the manuscript and being motivation for compiling our data for peer review – that measuring only few data points may have an influence on the g-value calculation. In that case the relative position of the prompt readout weighs particularly, especially if for better precision it is repeated several times.
- The fading test with the most intense preheat procedure of 60 s at 280 °C (**Fig. 4d–f**) still shows the stretching and finally updoming of the early part of the fading curve from aliquot 1 to aliquot 3. As many of the early values of the normalised signal are above one (overshooting for aliquot 3 up to ca. 1 – 1.5 decades) this does not allow reasonable g-value calculation for these fading curves. It seems that the electron redistribution lasts up to ca. 1.6 decades (here 11075 s or ca. 3 hours, respectively, in a test with 100 s laboratory irradiation and tc = 265 s) if stronger preheating procedures are applied. Or do we observe here another and/or additional effect?
- Although preheating after the delay time (prior to IRSL-readout; Rhodius et al. 2015) instead of preheating before the delay time (immediately after laboratory irradiation; Auclair et al. 2003) reduces the overall g-value, the fading curve still exhibits an initial (semi-)plateau ($T_{fad}$-16, **Fig. 2 j-l**). This seems to suggest that in addition to electron redistribution due to preheating (Auclair et al. 2003), which affects each data point in equal measure, other charge transfer processes could be responsible for the formation of an initial plateau and appear to be the dominating effect. Huntley & Lamothe (2001) argue that short recombination times would correspond to short distances between trap and recombination centers in the crystal lattice, which however become more and more unlikely with decreasing distance. Non-fading would be the result.

If the observation of an initial plateau is accepted, this would lead to the question of how to correctly handle the „prompt" readout. The position of the prompt readout may vary even for fading tests with equal laboratory irradiation times as, among others, the delay time for the earliest readout depends on the time of preheating and the time for reaching the liftup temperature. Therefore „prompt" is relative, but never immediate, and the data of the „prompt" readout is part of the (very early part of the) initial plateau, as observed in our measurements. In fact, it is the earliest measurable data point of the here detected (semi-)plateau, but not its origin.

Therefore the question arises: If the geologically relevant fading meachnism does not act on short delay times, is it correct to include the prompt readouts in the g-value calculation? In practice, this procedure serves to define the g-value slope most precisely close to the point of origin, but does it also define it accurately? Or do we get a higher precision at the expense of a less correct result?

If the „prompt" readout occured immediately after the laboratory irradiation or the preheating, one could possibly argue that electron redistribution has not yet fully started and therefore may possibly be neglected. But comparing tc with the length of time of the laboratory irradiation (half the time according to Auclair et al. 2003) plus the time for preheating shows that this assumption is not valid. Also, our fading curves show that the „prompt" dose points are part of the initial (semi-)plateau – although there are cases in which electron redistribution may increase (normalized IRSL signals > 1) for short but longer-than-prompt delay times.

[Figure]

**Fig 3** Results for HDS-713 on DA20 (Athenaeum) with preheat 60 s at 250 °C, IRSL readout at 50 °C, liftup temperature 50 °C. Laboratory dose 10.3 Gy (100 s beta irradiation time) and normalisation dose 5.2 Gy (50 s beta irradiation time). Graphs arbitrarily supplemented with g-values for parts of the fading curves and for selected data points.

- **(a – c)** only 120 s nitrogen flow at the beginning ($T_{fad}$-13)
- **(d – f)** continuous nitrogen flow ($T_{fad}$-14)
- **(g – i)** each aliquot measured separately in three individual sequences ($T_{fad}$-15)
- **(j – l)** pause before preheat (Rhodius et al. 2015) ($T_{fad}$-16)

[Figure]

**Fig 4** Results for HDS-713 on DA20 (Athenaeum) with IRSL readout at 60 °C and liftup temperature 60 °C. Laboratory dose 10.3 Gy (100 s beta irradiation time) and normalisation dose 5.2 Gy (50 s beta irradiation time). Graphs arbitrarily supplemented with g-values for parts of the fading curves and for selected data points.

- **(a – c)** Preheat 20 s at 280 °C ($T_{fad}$-17) *versus* **(d – f)** preheat 60 s at 280 °C ($T_{fad}$-18).

**4 pIR$_{1st}$IR$_{2nd}$-tests on polymineral fine grains and feldspar coarse grains**

For our study we had chosen polymineral fine grains assuming that potential inter-aliquot heterogeneity which may occur with coarse grains can be excluded for fine grains with several $10^5$ grains per aliquot. Further, not only fine grains but coarse-grain separates, too, contain different kinds of feldspar as (1) in practice sample preparation is not specific enough and (2) individual feldspar grains exhibit phase-exsolution lamellae. Nevertheless, we considered it worth investigating the reviewer's idea that our observations could be a specification of our fine grains, which are irrelevant for coarse grains.

**4.1 Methodical details**

We performed a fading test on three aliquots of feldspar coarse grains (150–200 µm), using: Norfloat Potash Feldspar, G-40 Feldspar and F-20 Feldspar with likely potassium contents ($KO_2$) of 12.0 wt.-%, 10.4 wt.-% and 4.1 wt.-%, respectively (cf. **Table i** in the appendix). The material was only resieved but not further processed (e.g., no etching). The test was performed on another Risø luminescence reader model TL/OSL DA20 (No. 245; same specifications as the reader used for the IRSL tests on the polymineral fine grains) utilizing small aliquots with few $10^1$ grains each fixed with silicon spray (hole mask ø 1 mm) on aluminium cups (ø ca. 10 mm).

This time, however, for the fading test a post-IR IRSL approach (Thomsen et al. 2008) was applied using a preheat of 60 s at 280 °C, IRSL at 60 °C for 240 s and post-IR IRSL at 225 °C

for 240 s ($pIR_{60}IR_{225}$). The test was performed after one-time 2 minutes N-purge at the beginning.

For comparison with polymineral fine grains we also performed $pIR_{60}IR_{225}$-tests on another sample of the loess-borne sediments from SW-Germany (HDS-511; drilling core HBIII, 750 – 757 cm; Kadereit et al. 2011). These tests were performed with different modes of N use. Here we give an example of a measurement with repeated N-purge (2 minutes N-purge after each SAR cycle).

To compensate for the loss of intensity of the $IR_{225}$-signal the laboratory dose for the fine-grain test was increased (from 100 s for the IRSL tests) to 400 s. Such measure was not necessary for the coarse-grain samples, which for $IR_{225}$ showed an increase in signal intensity as compared to $IR_{60}$. The $pIR_{1st}IR_{2nd}$-tests of the fine grains were performed still under the erroneous assumption that a pause-input of 10 s in the sequence editor adds a delay-time of 10 s in the measurement. This is why these measurements, too, show an excess of prompt readouts (zero-values on the x-axis). Only for the $pIR_{60}IR_{225}$-test on the coarse grains the shorter delay times were elongated. In addition, the maximum delay time was enlarged to ca. 80 h, as compared to ca. 10 h for the IRSL tests and ca. 20 h for the $pIR_{60}IR_{225}$-tests on the fine grains.

The differing times of laboratory irradiation result in differing values for tc and the differing maximum delay times further modulate the period („decades") covered on the x-axis. Further, tc for $IR_{225}$ is always larger than tc for $IR_{60}$ of the same $pIR_{60}IR_{225}$-measurement, as the $IR_{60}$-readout (duration 240 s) precedes the $IR_{225}$-readout. This leads to a comparably shorter decade-coverage of $IR_{225}$ as comapred to $IR_{60}$. Details of the SAR protocols are given in **Table ii** of the appendix. These explain the variations in decade-coverage. However, these variations are not crucial for the overall shape of the fading curves of the $pIR_{60}IR_{225}$-tests, which are shown in **Fig. 5 – 6** (fine-grain sample HDS-511) and **Fig. 7 – 8** (coarse-grain feldspar samples).

**4.2    Results**

A a result

– The $IR_{60}$ readouts of the coarse grain tests show few outlier data points (marked with red circles in **Fig. 7**) which, however, are not crucial for the issues adressed in the following.
– Most fading curves exhibit an initial part with a lower gradient in data values followed by a section with a stronger gradient. The data values of the initial part may scatter around 1 or even exceed this threshold value of the first measured data point. Values above 1 do not conform to the model of anomalous signal fading. In other cases, the initial plateaus or ridges are less well defined, but may show up as flat-stepped stairs. In this respect, $IR_{1st}IR_{2nd}$ measurements resemble IRSL measurements.
– $IR_{60}$-(semi-)plateaus appear shorter and/or less pronounced than their IRSL counterparts. This may be explained by the comparably larger tc-values. Additionally, this may result from stronger optical and thermal washing by the $IR_{225}$-readout accomplishing each SAR cycle.

[Figure]

**Fig. 5** pIR$_{60}$IR$_{225}$-fading measurement on polymineral fine-grain sample HDS-511 – here IRSL at 60 °C. SAR measurement with 400 s beta irradiation, 60 s preheating at 280 °C, pause, IRSL at 60 °C for 240 s and IRSL at 225 °C for 240 s. Repeated N-purge: 2 min N-purge at the end of each SAR cycle. Time since irradiation (log$_{10}$[t/tc]) on a logarithmic scale with zero-values not presentable (**left**) and on a linear scale (**right**). tc is 584 s. All g-values normalised to 2 days. Longest delay time ca. 20 h.

[Figure]

**Fig. 6** pIR$_{60}$IR$_{225}$-fading measurement on polymineral fine-grain sample HDS-511 – here IRSL at 225 °C. SAR measurement with 400 s beta irradiation, 60 s preheating at 280 °C, pause, IRSL at 60 °C for 240 s and IRSL at 225 °C for 240 s. Repeated N-purge: 2 min N-purge at the end of each SAR cycle. Time since irradiation (log$_{10}$[t/tc]) on a logarithmic scale with zero-values not presentable (**left**) and on a linear scale (**right**). tc is 893 s. All g-values normalised to 2 days. Longest delay time ca. 20 h.

[Figure]

**Fig. 7** pIR$_{60}$IR$_{225}$-fading measurement on coarse-grain (150 – 200 µm) feldspars – here IRSL at 60 °C. SAR measurement with 200 s beta irradiation, 60 s preheating at 280 °C, pause, IRSL at 60 °C for 240 s and IRSL at 225 °C for 240 s. 2 min N-purge at the start of the measurement. Time since irradiaton [log$_{10}$(t/tc)] on a logarithmic scale with zero-values not presentable **(left)** and on a linear x-scale (**right**). Red circles denoting outliers. tc is 584 s. All g-values normalised to 2 days. Maximum delay time ca. 80 h. **(b)** Olive line signature and numbers in brackets g-value without very first outlier data point.

[Figure]

**Fig. 8** $pIR_{60}IR_{225}$-fading measurement on coarse-grain (150 – 200 µm) feldspars – here IRSL at 225 °C. SAR measurement with 200 s beta irradiation, 60 s preheating at 280 °C, pause, IRSL at 60 °C for 240 s and IRSL at 225 °C for 240 s. 2 min N-purge at the start of the measurement. Time since irradiaton [$\log_{10}(t/tc)$] on a logarithmic scale with zero-values not presentable (**left**) and on a linear x-scale (**right**). tc is 692 s. All g-values normalised to 2 days. Maximum delay time ca. 80 h.

– IR$_{225}$-(semi-)plateaus are longer than the IR$_{60}$-(semi-)plateaus, which may be caused by thermal and IR stimulation by the preceding IR$_{60}$-readout.

– Next to an initial plateau IR$_{225}$-readouts can show a rather flat gradient also for longer delay times (**Fig. 7b & 7f**). For the potash feldspar the g-value for the complete data set is still around zero, as the difference in level between the initial plateau and the later values is minimal. For the F-20 feldspar, however, for which the initial plateau ends quite abruptly

the difference is much larger. Taking into account data points representing both levels leads to an erroneously large g-value. Therefore, this example seems to illustrate how measurement artifacts for $IR_{2nd}$-g-values can be produced. Whether the gradient of the data values representing longer delay times of the G-40 feldspar (**Fig. 7d**) represents the fading rate, or whether (in part) it is overprinted by electron redistribution could perhaps be clarified with the measurement of longer delay times. It appears that the shape of a fading curve is dependent on the degree to which the electron redistribution plateau reaches above the later part of the fading curve (little for the potash feldspar, noticeable for the F-20 feldspar) and whether the initial plateau ends more abruptly (F-20 feldspar) or expires more gradually (perhaps G-40 feldspar?).

**Conclusion**

Our fading measurements with unusually short delay times often exhibit an initial part of the fading curve with a comparably small gradient, often with g-values around 0.

We observed initial (semi-)plateaus for IRSL at 60 °C as well as $IR_{60}$ and $IR_{225}$ in the frame of an $IR_{1st}IR_{2nd}$-SAR protocol. The length of the initial (semi-)plateau proofed comparably longer for $IR_{2nd}$ than for $IR_{1st}$, likely promoted by electron excitation through the IR- and thermal stimulation of the preceeding $IR_{1st}$-readout.

Our earlier observations of an initial (semi-)plateau in the data curves of IRSL SAR fading tests on polymineral fine grains (as shown in the manuscript) apply also to $pIR_{1st}IR_{2nd}$ SAR fading tests both on (1) polymineral fine grains and (2) feldspar coarse grains (as shown in the final response). The latter allow an insight into how g-value artifacts may be generated for $IR_{2nd}$.

Several reasons are possible for an initial plateau in the fading curve: Tunneling afterglow or tunneling luminescence was observed after laboratory irradiation (Visosekas 1985, 1993; Molodkov et al. 2007). Huntley & Lamothe (2001) argue that the fading model based on tunneling of trapped electrons to nearby recombination centers does not apply for very short delay times after irradiation, due to the discrete nature of the crystal lattice and the low probability of very short distances between trap and center which would correspond to very quick recombination. These explanations are consistent with the occurance of an initial plateau. The plateau could also be a result of electron redistribution due to preheating (Auclair et al. 2003). As, however, both the fading test variants with preheating immediately after laboratory irradiation and preheating immediately before IR-readout produced a plateau, another charge-transfer process likely aids the plateau generation. Electron band-tail hopping (Guérin and Visocekas, 2015) might be a relevant mechanism. The longer $IR_{225}$-plateau could also be descriptively explained by the use-up of nearby recombination centers through the preceding $IR_{60}$ stimulation. More distant electron-hole pairs then recombine only after prolonged delay times, in agreement with a longer initial plateau of the fading curve.

The occurance of an initial (semi-)plateau raises the question whether prompt readouts should be included in a g-value estimation and whether the first data point of a fading curve should be delayed sufficiently – after the end of a likely plateau. The inclusion of prompt readouts may

be especially disadvantageous for $IR_{2nd}$-g-values in the frame of $pIR_{1st}IR_{2nd}$ SAR fading tests and explain to some extent why $IR_{2nd}$-g-values can be erroneously large.

**References**

Aitken, M.J.: An Introduction to Optical Dating, Oxford University Press, London, 267 pp, 1998.

Auclair, M., Lamothe, M., Huot, S.: Measurement of anomalous fading for feldspar IRSL using SAR. Radiation Measurements, 37, 487–492, https://doi.org/10.1016/S1350-4487(03)00018-0, 2003.

Huntley, D.J., Lamothe, M.: Ubiquity of anomalous fading in K-feldspars and the measurement and correction for it in optical dating, Canadian Journal of Earth Sciences 38, 1093-1106, https://doi.org/10.1139/e01-013, 2001.

Kadereit, A., Kühn, P., Wagner, G.A.: Holocene relief and soil changes in loess-covered areas of south-western Germany: The pedosedimentary archives of Bretten-Bauerbach (Kraichgau), Quaternary International 222 (1-2), 96 – 119, https://doi.org/10.1016/j.quaint.2009.06.025, 2010.

Kreutzer, S., Burow, C.: analyse_FadingMeasurement(): Analyse fading measurements and returns the fading rate per decade (g-value). Function version 0.1.14. In: Kreutzer, S., Burow, C., Dietze, M., Fuchs, M.C., Schmidt, C., Fischer, M., Friedrich, J., 2020. Luminescence: Comprehensive Luminescence Dating Data Analysis. R package version 0.9.8.9000-17. https://CRAN.R-project.org/package=Luminescence, 2020.

Molodkov, A., Jaek, I., Vasilchenko, V.: Anomalous fading of IR-stimulated luminescence from feldspar minerals: Some results of the study, Geochronometria, 26, 11–17, https://doi.org/10.2478/v10003-007-0007-0, 2007.

Murray, A.S., Wintle, A.G.: Luminescence dating of quartz using an improved single aliquot regenerative-dose protocol, Radiation Measurements, 32, 57–73, https://doi.org/10.1016/S1350-4487(99)00253-X, 2000.

Rhodius, C., Kadereit, A., Siegel, U., Schmidt, K., Eichmann, R., Khalil, L.A.: Constraining the time of construction of the irrigation system of Tell Hujayrat al-Ghuzlan near Aqaba, Jordan, using high-resolution optically stimulated luminescence (HR-OSL) dating, Archaeological and Anthropological Sciences, 9 (3), 345 –370, https://doi:10.1007/s12520-015-0284-x, 2015 (first online).

Thiel, C., Buylaert, J.-P., Murray, A., Terhorst, B., Hofer, I., Tsukamoto, S., Frechen, M.: Luminescence dating of the Stratzing loess profile (Austria) – Testing the potential of an elevated temperature post-IR IRSL protocol, Quaternary International, 234, 23 – 31, https://doi.org/10.1016/j.quaint.2010.05.018, 2011.

Thomsen, K.J., Murray, A.S., Jain, M., Bøtter-Jensen, L.: Laboratory fading rates of various luminescence signals from feldspar-rich sediment extracts, Radiation Measurements, 43, 1474–1486, https://doi.org/10.1016/j.radmeas.2008.06.002, 2008.

Visocekas, R.: Tunnelling radiative recombination in labradorite: its association with anomalous fading of thermoluminescence, Nuclear Tracks and Radiation Measurements, 10 (4–6), 521–529, https://doi.org/10.1016/0735-245X(85)90053-5, 1985.

Visocekas, R.: Tunneling radiative recombination in K-feldspar sanidine, Nuclear Tracks and Radiation Measurements, 21, 175–178, https://doi.org/10.1016/1359-0189(93)90073-I, 1993.

**Table i** Coarse-grain feldspar used for a $pIR_{60}IR_{225}$ SAR fading test and analytical data from the DigitalFire.com Reference Library.

| Material | Grain-size fraction | Sieved again | $K_2O$ [weight-%] | $Na_2O$ [weight-%] | CaO [weight-%] |
|---|---|---|---|---|---|
| Norfloat Potash Feldspar 371214 (Cookson Matthey) | 150 µm < x < 200 µm | Yes (CM) | 12.0 | 2.9 | 0.4 |
| G-40 Feldspar (Feldspar Corporation) | 150 µm < x < 200 µm | Yes (CM) | 10.4 | 3.0 | 0.8 |
| F-20 Feldspar (Feldspar Corporation) | 150 µm < x < 200 µm | Yes (JA) | 4.1 | 6.82 | 1.4 |

- Source of information: DigitalFire.Com Reference Library:
- G-40 Feldspar: https://digitalfire.com/4sight/material/g-40_feldspar_801.html
- F-20: https://digitalfire.com/material/f-20+feldspar
- Norfloat Feldspar: https://digitalfire.com/material/norfloat+feldspar

**Table ii** Specifications of the IRSL- and pR$_{60}$IR$_{225}$-SAR protocols. Measurements on luminescence readers model Risø TL/OSL DA20: IRSL measurements on reader no. 240 („Athenaeum), pR$_{60}$IR$_{225}$ measurements on reader no. 245.

| SAR protocol step | | material | polymineral fine grains | polymineral fine grains | feldspar coarse grains |
|---|---|---|---|---|---|
| | | protocol | SAR IRSL | SAR pIR$_{60}$IR$_{225}$ | SAR pIR$_{60}$IR$_{225}$ |
| | | sample | HDS-713 | HDS-511 | Potash feldspar, G-40, F-20 |
| [1] | | laboratory dose | ß-IRR 100 s | ß-IRR 400 s | ß-IRR 200 s |
| [2] | or
or | preheat procedure | PHT 60 s at 250 °C (T$_{fad}$ -13 to T$_{fad}$ -16)
PHT 20 s at 280 °C (T$_{fad}$ -17)
PHT 60 s at 280 °C (T$_{fad}$ -18) | PHT 60 s at 280 °C | PHT 60 s at 280 °C |
| [3] | | prompt and delayed readouts | 3 x prompt, …, max. ca. 10 h, 3 x prompt | 3 x prompt, …, max. ca. 20 h, 3 x prompt | 3 x prompt, …, max. ca. 80 h, 3 x prompt |
| [4a]

[4b] | or | IRSL-readout | IRSL 240 s at 50 °C (T$_{fad}$ -13 to T$_{fad}$ -16)
IRSL 240 s at 60 °C (T$_{fad}$ -17 to T$_{fad}$ -18) | IRSL 240 s at 60 °C

IRSL 240 s at 225 °C | IRSL 240 s at 60 °C

IRSL 240 s at 225 °C |
| [5] | | normalisation dose (test dose) | ß-IRR 50 s | ß-IRR 200 s | ß-IRR 100 s |
| [6] | or
or | preheat procedure | PHT 60 s at 250 °C (T$_{fad}$ -13 to T$_{fad}$ -16)
PHT 20 s at 280 °C (T$_{fad}$ -17)
PHT 60 s at 280 °C (T$_{fad}$ -18) | PHT 60 s at 280 °C | PHT 60 s at 280 °C |
| [7a]
[7b] | | IRSL-readout | IRSL 240 s at 60 °C | IRSL 240 s at 60 °C
IRSL 240 s at 225 °C | IRSL 240 s at 60 °C
IRSL 240 s at 225 °C |
| | | tc | tc$_{IRSL}$  279 s (T$_{fad}$ -13, -15), 263 (T$_{fad}$ -14),
280 - 281 (T$_{fad}$ -16), 223-224 (T$_{fad}$ -17),
265 s (T$_{fad}$-18) | tc$_{IR60}$  583 - 584 s
tc$_{IR225}$  892 - 893 s | tc$_{IR60}$  383-384 s
tc$_{IR225}$  692 s |